# Targeting *CCNE1* amplified ovarian and endometrial cancers by combined inhibition of PKMYT1 and ATR

Haineng Xu [1,2,8], Erin George[1,2,6,8], David Gallo[3,7,8], Sergey Medvedev[1,2], Xiaolei Wang[1,2], Arindam Datta[4], Rosie Kryczka[3], Marc L. Hyer[5], Jimmy Fourtounis[3], Rino Stocco[3], Elia Aguado-Fraile[5], Adam Petrone[5], Shou Yun Yin[3], Ariya Shiwram[3], Fang Liu[1,2], Matthew Anderson[1,2], Hyoung Kim [1,2], Roger A. Greenberg [4], C. Gary Marshall[5] & Fiona Simpkins [1,2] ✉

Ovarian cancers (OVCAs) and endometrial cancers (EMCAs) with *CCNE1*-amplification are often resistant to standard treatment and represent an unmet clinical need. Synthetic-lethal screening identified loss of the CDK1 regulator, PKMYT1, as synthetically lethal with *CCNE1*-amplification. We hypothesize that *CCNE1*-amplification associated replication stress will be more effectively targeted by combining PKMYT1 inhibitor lunresertib (RP-6306), with ATR inhibitor camonsertib (RP-3500/RG6526). Low dose combination RP-6306 with RP-3500 synergistically increases cytotoxicity more so in *CCNE1*-amplified compared to non-amplified cells. Combination treatment produces durable antitumor activity, reduces metastasis and increases survival in *CCNE1*-amplified patient-derived OVCA and EMCA xenografts. Mechanistically, low doses of RP-6306 with RP-3500 increase CDK1 activation more so than monotherapy, triggering rapid and robust induction of premature mitosis, DNA damage, and apoptosis in a *CCNE1*-dependent manner. These findings suggest that targeting CDK1 activity by combining RP-6306 with RP-3500 is an effective therapeutic approach to treat *CCNE1*-amplifed OVCAs and EMCAs.

Despite significant progress in the last twenty years in ovarian cancer (OVCA) treatment and improved mortality rates[1], this disease remains the most lethal gynecologic malignancy[2]. Such progress has particularly favored patients with germline or somatic *BRCA1/2* pathogenic mutations or those with tumors exhibiting homologous recombination (HR) deficiency[3,4]. On the other hand, patients with HR-proficient tumors, particularly those with *CCNE1* gene amplification, exhibit de novo or rapid emergence of chemotherapy resistance and poor survival[5,6]. Endometrial cancer subtypes share molecular profiles with HGSOC, such as *CCNE1* amplification and *TP53* mutations. There is

[1]Penn Ovarian Cancer Research Center, Perelman School of Medicine, University of Pennsylvania, Philadelphia, PA, USA. [2]Department of Obstetrics and Gynecology, Division of Gynecologic Oncology, Hospital of the University of Pennsylvania, Philadelphia, PA, USA. [3]Repare Therapeutics, Inc., 7171 Frederick-Banting, Ville St-Laurent, QC, Canada. [4]Department of Cancer Biology, Penn Center for Genome Integrity, Basser Center for BRCA, Perelman School of Medicine, University of Pennsylvania, Philadelphia, PA, USA. [5]Repare Therapeutics, Inc., 101 Main St, Cambridge, MA, USA. [6]Present address: Department of Gynecologic Oncology, H. Lee Moffitt Cancer Center and Research Institute, Tampa, FL, USA. [7]Present address: Department of Medicine, University of Wisconsin-Madison, Madison, WI, USA. [8]These authors contributed equally: Haineng Xu, Erin George, David Gallo. ✉e-mail: fiona.simpkins@pennmedicine.upenn.edu

an increasing incidence in high-risk EMCA histologic subtypes such as uterine serous carcinoma and carcinosarcoma[7], and these subtypes are among the cancers with the highest incidence of *CCNE1* gene amplification[8]. Further, mortality rates for endometrial cancer (EMCA) are overall increasing, and the two-fold higher risk of death from ovarian cancer compared to endometrial cancer in the early 1990s has virtually been eliminated by oppositional mortality trends[1]. To date, there are no FDA-approved drugs for *CCNE1* amplified cancers, including OVCA and EMCA, despite preclinical data showing *CCNE1* amplification is targetable therapeutically[9–12]. Given this unmet clinical need, we sought to identify a treatment strategy that targets critical survival pathways for *CCNE1*-amplified dependent gynecological cancers.

Cyclin E1 binds and activates CDK2, a key regulatory element that promotes initiation of DNA replication and G1/S cell cycle progression[13,14]. *CCNE1* amplification and cyclin E1 overexpression prematurely activate CDK2 and initiation of DNA replication before cells have time to sufficiently license pre-replication complexes at origins[15]. Unscheduled origin firing leads to replication stress and ensuing DNA damage and genome instability[9,11,16]. Cyclin E-CDK2 complexes directly activate the MYBL2-MuvB-FOXM1 (MMB) transcriptional network causing early activation of the G2/M transcriptional network and accumulation of cytoplasmic cyclin B in S-phase[17]. To slow the onset of mitosis and allow time to compete faithful genome duplication, *CCNE1* amplified cells activate the S and G2/M cell cycle checkpoints to suppress cell cycle transitions, allowing time to complete DNA replication[11]. We reasoned that increased dependence on S and G2/M checkpoints for survival in *CCNE1* amplified cells represents a therapeutic vulnerability which is the focus of this study.

Membrane-associated tyrosine- and threonine-specific Cdc2-inhibitory kinase (PKMYT1) and WEE1 kinase are each critical G2/M checkpoint regulators that inhibit cell cycle progression by catalyzing inhibitory phosphorylation of CDKs[18–21]. WEE1 restricts both CDK1 and CDK2 activity by phosphorylating Tyr15, while PKMYT1 selectively restricts CDK1 activity by phosphorylating Thr14[22]. WEE1 is localized to the nucleus, while PKMYT1 is tethered to the cytoplasmic face of the ER/Golgi, where it physically interacts with cyclin B[21,23]. PKMYT1 and WEE1 inhibitors (PKMYT1i and WEE1i, respectively) are each reported to show preclinical efficacy as monotherapy in *CCNE1*-amplified and cyclin E overexpressing models by activating CDK1, which forces cells into mitosis prematurely with under replicated and damaged DNA[17,24,25].

Replication stress caused by *CCNE1* amplification also activates the ataxia telangiectasia and Rad3-related (ATR) kinase to promote the DNA damage response (DDR), stabilizing DNA replication forks from collapse into DNA double-strand breaks and preventing dormant origin activation[11]. ATR also activates CHK1, that limits mitotic progression by phosphorylating and inhibiting the CDC25 family phosphatases required for dephosphorylation and activation of CDK1[26]. *CCNE1* amplification also sensitizes cells to ATR inhibition[27].

Previous studies indicate that targeting S and G2/M checkpoint kinases selectively kill *CCNE1*-amplified and overexpressing cells. WEE1 inhibitors (WEE1i) are reported to show preclinical efficacy as monotherapy in *CCNE1*-amplified and overexpressing models[17,24,25]. In certain contexts, ATR inhibitors (ATRi) are also reported to show cytotoxic activity in *CCNE1* overexpressing models with high levels of replication stress[27]. We have previously reported that low doses of WEE1i and ATRi synergize in *CCNE1*-amplified OVCA and EMCA preclinical models by increasing replication fork collapse and DNA damage[11]. Intriguingly, a recent CRISPR-Cas9 genome-wide screen using an isogenic pair of cell lines that stably overexpress cyclin E from a *CCNE1-2A-GFP* fusion integrated into the genome of RPE1-hTERT *TP53*[−/−] Cas9 cells identified PKMYT1 as a top hit for synthetic lethality in the *CCNE1* overexpressing cells. This study further showed that PKMYT1 inhibition (PKMYT1i) alone induces cell death by forcing cells from the S-phase into mitosis

with under-replicated DNA, resulting in mitotic catastrophe[17]. Given *CCNE1*-amplified cells rely on the G2/M checkpoint to attenuate lethal levels of replication stress-induced DNA damage by upregulating the ATR axis for DNA repair[24], we hypothesized that dual inhibition of PKMYT1 and ATR (PKMYT1i-ATRi) will further enhance CDK1 activation and cytotoxicity, especially towards *CCNE1* amplified or over-expressing tumor cells allowing lower dosing strategies and potentially alleviate toxicity.

There is currently one PKMYT1i and several ATRi in clinical development. Lunresertib (RP-6306), is a novel, first in class, selective and orally bioavailable PKMYT1i[28]. RP-6306 is currently in clinical development as monotherapy or in combination with chemotherapy or targeted therapies (e.g., camonsertib) for the treatment of patients with solid tumors harboring *CCNE1* amplification or *FBXW7/PPP2R1A* inactivating mutations (NCT04855656, NCT06107868, NCT05147272)[29]. There are currently seven ATR inhibitors in clinical development (RP-3500, AZD6738, BAY1895344, ATRN119, M1774, ART0380, IMP9064; NCT04497116, NCT01955668, NCT03188965, NCT04905914, NCT04170153, NCT04657068, NCT05269316). Camonsertib (RP-3500) is a potent, selective, and orally bioavailable ATR inhibitor currently in clinical development as monotherapy and in combination with other targeted therapies, including PARPi, for the treatment of patients with solid tumors harboring selected DDR alterations, including *BRCA1/2* or *ATM* inactivating mutations (NCT04972110, NCT05405309)[30,31].

In this study, we report that combining RP-6306 with RP-3500 (PKMYT1i-ATRi) increases CCNE1 level-dependent cytotoxicity in a large panel of OVCA and EMCA cell lines. The increased cytotoxicity translates to significantly improved activity and overall survival in mice bearing *CCNE1*-amplified cell lines and patient-derived xenografts. Increased cytotoxicity and activity is attributed to synergistic CDK1 activation in the S-phase of *CCNE1* overexpressing or amplified cells triggering unscheduled mitosis before completion of DNA replication, leading to lethal levels of DNA damage and increased apoptosis. We find that *CCNE1* amplification is a robust biomarker predictive of sensitivity towards low doses of PKMYT1i-ATRi, with limited effects observed in *CCNE1*[LOW] or *BRCA1/ATM* mutated cells. Taken together our studies support the clinical development of PKMYT1i-ATRi combinations for treatment of tumors with *CCNE1* amplification and/or overexpression.

## Results

### Combination PKMYT1i-ATRi is synergistic in *CCNE1* amplified OVCA and EMCA cells

To investigate the dependence of PKMYT1i on CCNE1 level in OVCA cells (OVCAR3, FUOV1, COV318, OVCAR8, OVSAHO, Kuramochi, WO-20, SKOV3) and EMCA cells (KLE, MFE280, SNU685), RP-6306 efficacy in cancer cells with *CCNE1* amplification (*CCNE1*[AMP]), *CCNE1* copy gain (*CCNE1*[GAIN]), or *CCNE1* neutral (*CCNE1*[LOW]) was investigated. RP-6306 was most active in *CCNE1*[AMP] cells, with reduced activity in *CCNE1*[GAIN] cells and very limited effect in *CCNE1*[LOW] cells, as evident by the difference in median IC$_{50}$ between these groups (Fig. 1A, B and Supplementary Fig. S1A). Since PKMYT1 inhibition can force cells into unscheduled mitosis by preventing CDK1 phosphorylation, we hypothesized that this effect could be enhanced by activation of CDC25 via inhibition of ATR. We therefore tested the ATR inhibitor (ATRi), RP-3500, as monotherapy and in combination with RP-6306. Similar to RP-6306, high CCNE1 expression levels did enrich for RP-3500 single-agent activity, as evident by the difference in median IC$_{50}$ between these groups (Fig. 1A, B and Supplementary Fig. S1A).

Combination of RP-6306 with RP-3500 showed a much stronger inhibitory effect than the respective monotherapy treatments, with significant synergistic effect by coefficient of drug interaction (CDI) in *CCNE1*[AMP] cells, less so in *CCNE1*[GAIN] cells, and very limited to no effect in *CCNE1*[LOW] cells (Fig. 1C). *CCNE1* copy number (CN) significantly

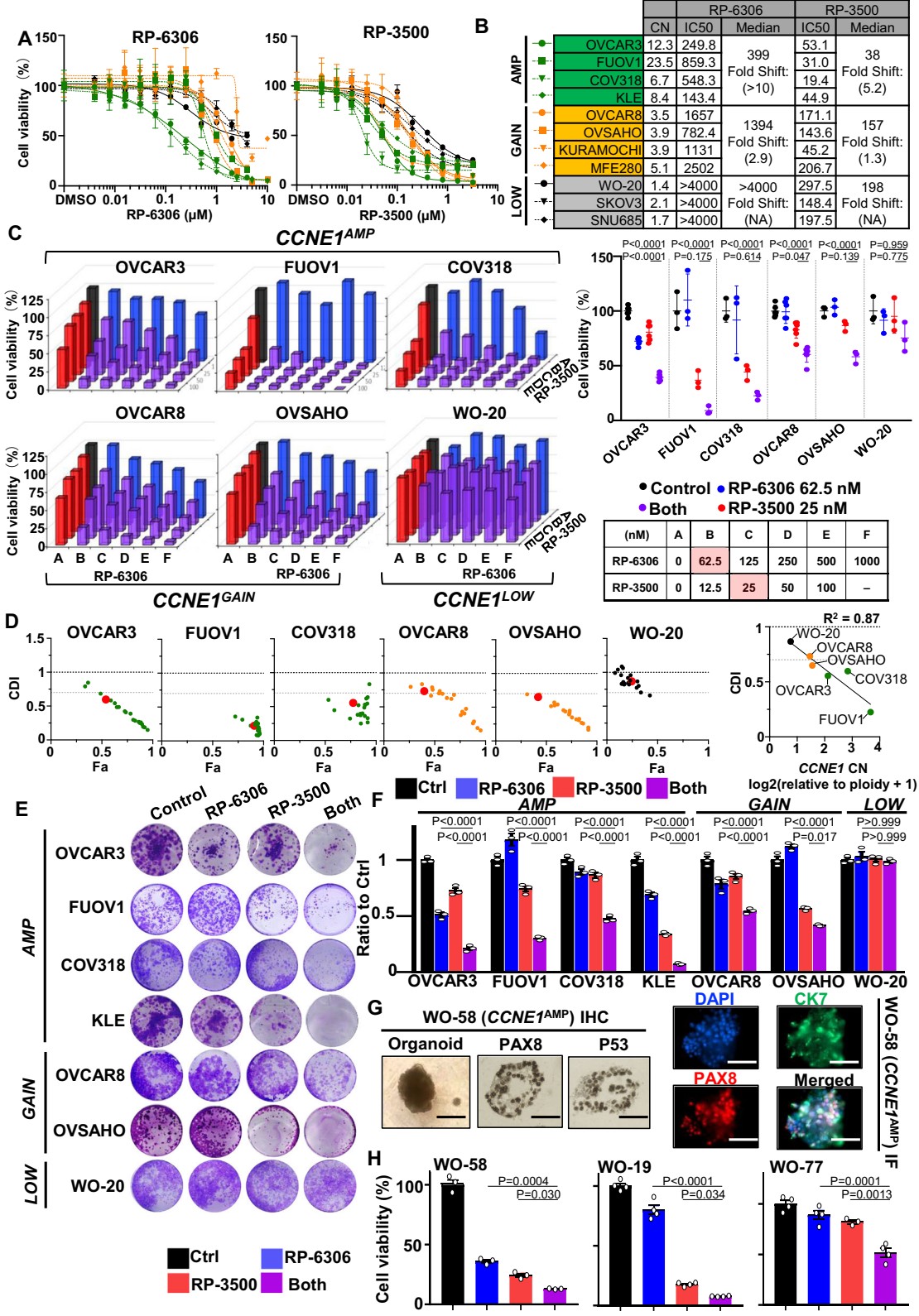

correlated with CDI ($R^2 = 0.87$, P = 0.0064 Fig. 1D). Fraction affected (Fa) cells and CDI calculation clearly illustrates that the synergistic effect of the combination decreases viability in a *CCNE1*-dependent manner by lower CDI values with higher fractions of cells affected (Fig. 1D). Colony formation assays yielded similar results, with the combination of RP-6306 and RP-3500 significantly inhibiting colony formation in *CCNE1*[AMP] cells, somewhat less effectively in *CCNE1*[GAIN]

cells, with minimal effects in *CCNE1*[LOW] cells (Fig. 1E-F, Figure S1B-C). In the cell lines investigated, we found *CCNE1* copy number correlated with mRNA and Cyclin E1 protein expression levels (Figure S1E).

To evaluate the efficacy of these compounds in a more clinically relevant models, we established patient-derived organoids from a *BRCA1* mutant, *CCNE1*[AMP] (CN = 7) homologous recombination (HR) proficient PDX model with acquired PARP inhibitor resistance,

**Fig. 1 | Combination PKMYT1i-ATRi is synergistic in CCNE1 amplified OVCA and EMCA cells. A, B** Detection of cells response to monotherapy of PKMYT1i, RP6306 (**A**) and ATRi, RP-3500 (**B**) with MTT assay. *CCNE1*[AMP] in blue, *CCNE1*[GAIN] in orange, and *CCNE1*[LOW] in black. Absolute CN and IC$_{50}$, and median IC$_{50}$ for all *CCNE1*[AMP], *CCNE1*[GAIN], and *CCNE1*[LOW] cell lines with fold shift relative *CCNE1*[LOW] lines (table). **C** Cell viability analysis of the indicated *CCNE1* amplified, gain, and low/neutral cell lines after treatment at indicated doses. Monotherapy for PKMYT1i, RP6306, is highlighted in blue, and ATRi, RP3500, is highlighted in red. Combinations are highlighted in purple. Assays were normalized by doubling time such that cells doubled at least twice. *n* = 3; Mean was presented. The dose highlighted in red in the table (bottom right) was selected as the most synergistic dose in the SNU685 *CCNE1* inducible cell line (Fig. 2F) and compared across all cell lines tested with the combination. *n* = 3; Mean ± S.D. **D** Coefficient of drug interaction (CDI) relative to the fraction affected (Fa) plot of the indicated cell lines from the MTT assay was calculated for each dose combination. The red dot corresponds to the red highlighted dose combination in (**C**). *CCNE1*[AMP] in Green, *CCNE1*[GAIN] in orange, and *CCNE1*[LOW] in black. CDI < 1 synergy with CDI < 0.7 significant synergy, CDI = 1 additive, CDI > 1 antagonistic. Plot of CDI versus *CCNE1* copy number at indicated dose highlighted in red (right). R$^2$ value shown (*P* = 0.0064). **E** Colony formation Analysis of PKMYT1i-ATRi combination in *CCNE1* amplified OVCA and EMCA cells with RP-6306 (31.3 nM), RP-3500 (6.25 nM) or combination for 10 days. **F** Quantification of colony formation assay in (**E**). **G** WO-58 organoids were developed from *CCNE1* amplified, *BRCA1* mutant HGSOC WO-58 and characterized with ovarian cancer marker PAX8 and epithelial marker CK7 by immunofluorescence (IF) and immunohistochemistry (IHC). P53 expression was detected by IHC. Scale bar: 50 μm. **H** Cell viability detection of PKMYT1i-ATRi combination on *CCNE1* amplified HGSOC organoids. WO-58, WO-19, and WO-77 organoids were treated with RP-6306 (250 nM), RP-3500 (50 nM), or both for 10 days and measured with CCK8 assay. *n* = 3 for WO-58 organoids, *n* = 4 for WO-19 and WO-77 organoids; Mean + SD. Significance determined by two-way ANOVA followed by Tukey's multiple comparisons test for (**C**) and (**F**). Simple linear regression calculated for (**D**). One-way ANOVA for followed by Tukey's multiple comparisons test for (**H**).

designated WO-58, *CCNE1*[AMP] WO-19 (CN = 11-23) and WO-77 (CN = 9) models. The organoid model is characterized by P53 mutation (p53 overexpression), ovarian marker PAX8, and epithelial marker CK7 (Fig. 1G). The combination of RP-6306 and RP-3500 showed increased growth inhibition than the respective monotherapies in WO-58, WO-19 and WO-77 organoids (Fig. 1H and Supplementary Fig. S1D). Taken together, these results suggest that the PKMYT1i-ATRi combination is more effective in OVCA and EMCA with higher levels of *CCNE1* copy number.

### CCNE1 induction augments PKMYT1i-ATRi cytotoxic effects

To further investigate whether the PKMYT1i-ATRi combination is dependent on Cyclin E1 protein level, we established and then tested drug effects in immortalized fallopian tube cells and ovarian and endometrial cancer cell lines, with and without *CCNE1* overexpression (FT282 with *CCNE1* overexpression, WO-20 with inducible *CCNE1* and SNU685 with inducible *CCNE1*)[11]. Strong synergy was observed in both parental and *CCNE1* overexpressing FT282 cells (Fig. 2A). However, the concentration of RP-6306 required for synergy was about 60-fold lower, and for RP-3500 about 4-fold lower in *CCNE1* overexpressing cells (Fig. 2A). While 938 nM RP-6306 and 7.8 nM RP-3500 showed no effect in combination on parental FT282 cells, FT282-*CCNE1* overexpressing cells treated with only 31 nM RP-6306 and 7.8 nM RP-3500 showed almost 80% growth inhibition (Fig. 2B). In SNU685 cells, induction of *CCNE1* increased sensitivity to RP-6306 alone by 4.8-fold, but had a limited effect on sensitivity to RP-3500 (Fig. 2C, D and Supplementary Fig. S2A). The PKMYT1i-ATRi combination showed limited activity in SNU685 parental cells, but with *CCNE1* induction, the combination was much more effective and demonstrated synergy (Fig. 2E, F). The effect of *CCNE1* induction on enhancing combination sensitivity in the OVCA WO-20 cell line was similar to that observed in SNU685 (Fig. 2G). The PKMYT1i-ATRi combination also inhibited colony formation in WO-20 and SNU685 cells with *CCNE1* induction, but very limited effect was observed in matched Cyclin E1 low cells (Fig. 2H, I and Supplementary Fig. S2B). To test that RP-6306 and RP-3500 effects are not off target, we knocked down PKMYT1 and ATR genes in OVCAR3 cells. The combination of PKMYT1 and ATR siRNAs showed decrease in viability compared to single siRNA treatment (Supplementary Fig. S2C). These results demonstrate that combination inhibition of PKMYT1i-ATRi clearly relies on cyclin E1 expression level for a cytotoxic effect.

### PKMYT1i-ATRi causes tumor regression in CCNE1 amplified PDX models

To study the in vivo anti-tumor activity of the PKMYT1i-ATRi combination, three patient-derived xenograft (PDX) models with *CCNE1* amplification were utilized, including two OVCA (WO-19, WO-77) and one EMCA (WU-115) model (Supplementary Fig. S3A, B). First,

tolerability and anti-tumor activity were explored using RP-6306 and RP-3500 monotherapy in NSG mice, and mice bearing WO-19 xenograft tumors, respectively. Both RP-6306 and RP-3500 monotherapy were tolerated, as indicated by stable body weights at their maximum tolerated doses (MTDs), of 20 mg/kg for both compounds (Supplementary Fig. S4A). Both RP-6306 and RP-3500 monotherapy yielded limited single-agent anti-tumor activity in WO-19 PDX (Supplementary Fig. S4B), necessitating evaluation of the PKMYT1i-ATRi combination.

To first evaluate the tolerability of the combination, non-tumor-bearing NSG mice were treated with a combination of RP-6306 and RP-3500, using multiple dose levels, in order to identify doses for evaluation in a follow-up efficacy study using tumor-bearing mice. There were no treatment cessations required for the mice in the combination RP-6306 at 5 mg/kg or 10 mg/kg (Day 1–5) with RP-3500 using intermittent dosing at 5 mg/kg (Day 1–3). There were only two dose reductions in the combination RP-6306 at 10 mg/kg with RP-3500 at 5 mg/kg (Supplementary Fig. S4A), yielding tolerable doses for evaluation next in tumor-bearing efficacy studies. Next, mouse plasma pharmacokinetic studies were performed to evaluate drug-drug interactions and overall free fraction exposure. The free plasma concentrations of RP-6306 and RP-3500 were similar when dosed as single agents or when used in combination, suggesting no significant drug-drug interactions (Supplementary Fig. S4C), and exposures were at or above levels previously associated with meaningful target pathway engagement[28,29].

The combination was next evaluated in OVCA and EMCA PDX models. RP-6306 at 5 mg/kg or 10 mg/kg with RP-3500 (5 mg/kg) resulted in significant tumor regressions compared to RP-6306 monotherapy at 10 mg/kg (*P* = 0.0240, *P* = 0.0025, respectively) and RP-3500 monotherapy in the CCNE1 amp EMCA model (WU-115) model (*P* = 0.0032, *P* = 0.002, respectively, Fig. 3A and Supplementary Fig. S5A). There was over a 6-fold improvement in median overall survival compared to control and an approximate 3.5-fold improvement in median overall survival relative to monotherapies (Fig. 3A). Combination treatment also decreased metastasis (organs with metastasis) compared to monotherapy or control but this was non-significant in this overall not a highly metastatic model (Fig. 3A). Combination PKMYT1i-ATRi also increased tumor suppression in the *CCNE1*[AMP] OVCA WO-19 and WO-77 PDX models, which are resistant to standard-of-care platinum chemotherapy (Fig. 3B, C and Supplementary Fig. S5B, C)[11]. For WO-19, a combination of RP-6306 at 10 mg/kg with RP-3500 (5 mg/kg) significantly increased tumor suppression relative to RP-6306 at 10 mg/kg (*P* < 0.0001) and RP-3500 (*P* < 0.0001) monotherapy and improved median overall survival relative to monotherapy (*P* = 0.0007, *P* = 0.0006, respectively; Fig. 3B). Notably, there was a significant decrease in metastases relative to control with combination treatment (*P* = 0.0401, *P* = 0.0122 for combination with 5 mg/kg and 10 mg/kg RP-6306, respectively; Fig. 3B). For WO-77,

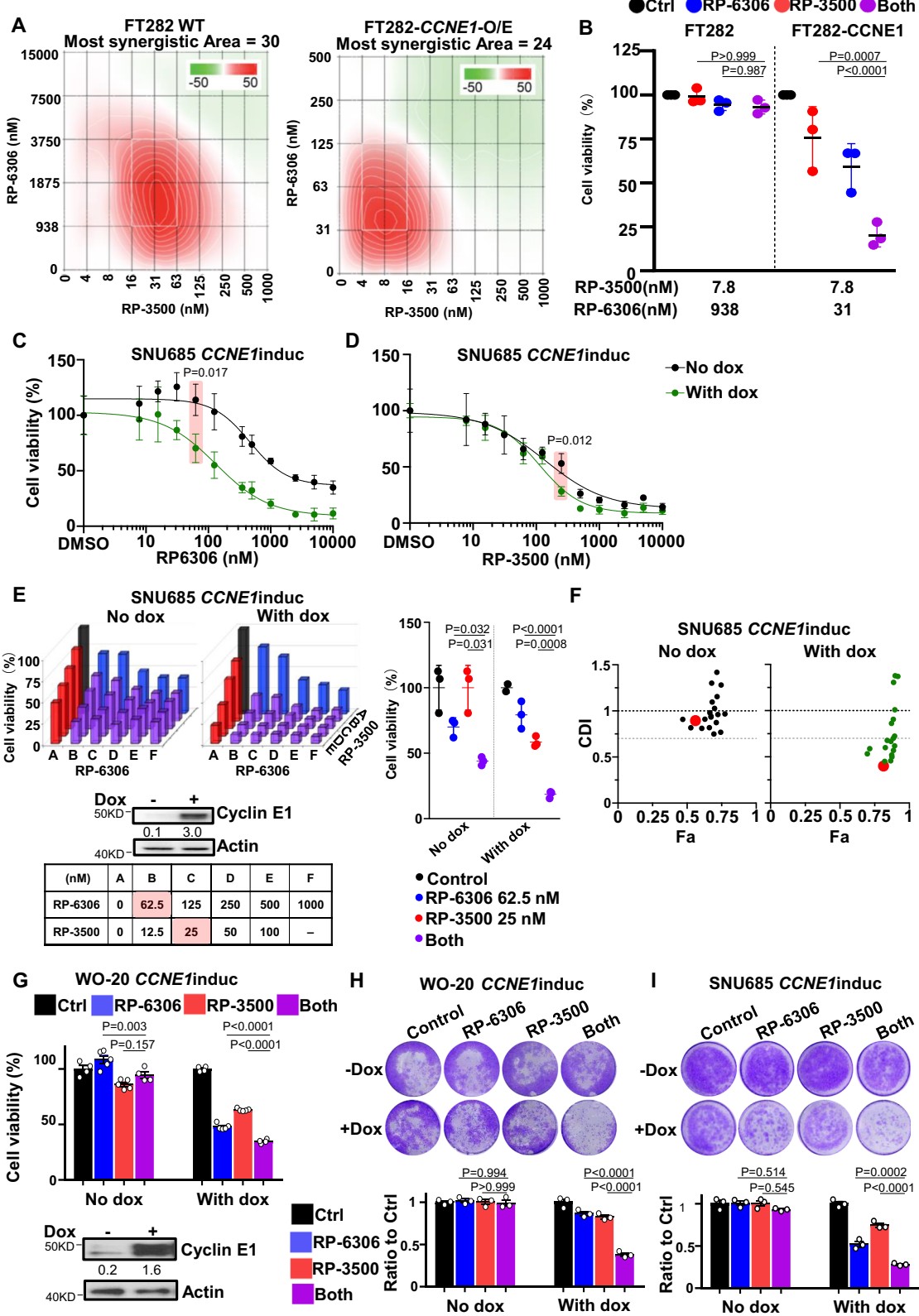

combination RP-6306 at 5 mg/kg with RP-3500 (5 mg/kg) significantly increased tumor suppression relative to RP-6306 at 5 mg/kg ($P < 0.0001$) and RP-3500 ($P = 0.0063$) monotherapy and improved median overall survival relative to monotherapy ($P < 0.0001$ and $P = 0.0003$, respectively). Also, there was a statistically significant decrease in metastases with combination treatment relative to control ($P = 0.0002$, $P < 0.0001$ for combination with 5 mg/kg and 10 mg/kg

RP-6306, respectively; Fig. 3C) and a significant decrease in ascites score relative to control ($P = 0.0261$, $P = 0.0102$, respectively, Supplementary Fig. S4D). Neither the WU-115 of WO-19 models made a significant amount of ascites (Supplementary Fig. S4D).

With all three PDX models, there was limited toxicity, as evident by body weight for both monotherapy and combination treatments (Fig. 3D). While the combined RP-6306/RP-3500 treatment groups

**Fig. 2 | Combination PKMYT1i-ATRi is synergistic in OVCA and EMCA cells depending on CCNE1 level. A** ZIP synergy scores at various dose combinations of RP-6306 and RP-3500 in FT282-hTERT *p53*[R175H] parental (WT) and CCNE1-overexpressing (*CCNE1*-O/E) cells. Score ≥10 (red color) represents synergy, ≤ − 10 (green) represents antagonism. Values were obtained by analyzing mean data from 3 independent biological replicates with SynergyFinder. **B** Growth inhibition relative to DMSO control of parental and CCNE1-overexpressing cells after treatment with the indicated dose of RP-6306, RP-3500, or the combination of both. *n* = 3; Mean + SD. **C, D** Cell viability detection of SNU685 cells in response to RP-6306 monotherapy (**C**) and RP-3500 monotherapy (**D**). *n* = 3; Mean ± SD. Highlighted dose showing the statistical difference in *CCNE1*-induced SNU685 cells with or without *CCNE1* induction. **E, F** Cell viability analysis of the indicated SNU685 *CCNE1* inducible cells ± doxycycline lines after treatment at the indicated doses. Doxycycline: 1 μg/ml. Monotherapy for PKMYT1i, RP6306, is highlighted in blue, for ATRi,

RP3500, is highlighted in red, Combinations are highlighted in purple for PKMYT1i-ATRi. Assays were normalized by doubling time such that cells doubled at least twice. *n* = 3; Mean ± SD. Growth inhibition relative to DMSO control of parental and *CCNE1*-overexpressing cells after treatment with indicated doses in pink (middle). The coefficient of drug interaction (CDI) relative to the fraction affected (Fa) plot of the indicated cell lines is indicated to the right of each bar graph. CDI < 1 synergy with CDI < 0.7 significant synergy, CDI = 1 additive, CDI > 1 antagonistic. The red dot corresponds to doses highlighted in pink. **G** Measurement of drug combinations in WO-20 *CCNE1^inducible* cells with or without Cyclin E1 induction. *n* = 4; Mean + SD. **H, I** Colony formation analysis (Upper panels) and quantification (Lower panels) of WO-20 *CCNE1^inducible* cells (**H**) and SNU685 *CCNE1^inducible* (**I**) in response to RP-6306 (31.3 nM), RP-3500 (6.25 nM) or combination for 10 days. Significance determined by two-way ANOVA followed by Tukey's multiple comparisons test for (**B, E, G**) and two-way ANOVA followed by Tukey's multiple comparisons for (**C, D**).

exhibit significant tumor regressions and improved overall survival compared to vehicle and monotherapy groups, we eventually observe tumor outgrowth after several weeks of treatment. We believe this is likely due to acquired resistance mechanisms such as loss of *CCNE1* CN or downregulation of MMB-FOXM1 transcription[17]. We also tested the combination PKMYT1i-ATRi in the *CCNE1*^AMP OVCAR3 cell line xenograft and found superior tumor growth inhibition (TGI) with the combination compared to RP-6306 and RP-3500 monotherapy (P < 0.001) and showed limited toxicity (Fig. 3E and Supplementary Fig. S5D). These data collectively demonstrate that combination inhibition of PKMYT1 and ATR significantly suppresses tumor growth and improves median overall survival in *CCNE1*^AMP OVCA and EMCA.

## ATRi enhances DNA damage and premature mitosis caused by PKMYT1i

We investigated if a combination PKMYT1i-ATRi treatment causes defective cell cycle progression and DNA damage leading to cell death. To measure cell cycle progression and DNA damage, we used quantitative imaging-based cytometry (QIBC) with 5-ethynyl-20 -deoxyuridine (EdU) incorporation to label S-phase cells, 4′,6-diamidino-2-phenylindole (DAPI) to measure DNA content and γH2AX as a marker of DNA damage after PKMYT1i-ATRi treatment using FT282 *CCNE1*-overexpressing cells (Fig. 4A and Supplementary Fig. S6A, B). The combination of RP-6306 with RP-3500 led to a substantial increase in DNA damage in *CCNE1* overexpressing FT282 cells based on the appearance of pan-γH2AX-positive (pan-γH2AX⁺) cells compared to parental cells (Fig. 4A). Most of the pan-γH2AX⁺ cells had <2 C DNA content and were EdU⁻, suggesting the cells with DNA damage were unable to complete DNA replication (Supplementary Fig. S6A). Consistent with a defect in DNA replication, the proportion of EdU⁺ *CCNE1*-overexpressing cells was dramatically decreased when treated with both RP-6306 and RP-3500 compared to RP-6306 or RP-3500 alone (Supplementary Fig. S7C). Notably, the most dramatic increase of DNA damage in *CCNE1*-overexpressing cells was observed at RP-6306 and RP-3500 concentrations that have little effect as a single agent or in parental FT282 cells (Fig. 4A). Combination RP-6306 with RP-3500 treatment increased γH2AX levels in *CCNE1*^AMP (OVCAR3, KLE, and FUOV1) and *CCNE1*^GAIN (OVCAR8) but not *CCNE1*^LOW (WO-20) cells indicating that tumors with *CCNE1* amplification are susceptible to DNA damage induced by combination PKMYT1i-ATRi (Fig. 4B–D). Combination of very low dosage RP-6306 (31.3 nM) with RP-3500 at (6.25 nM) also show similar trend in increasing γH2AX levels in a time-dependent manner (Supplementary Fig. S6D, E).

To further probe the source of DNA damage we measured micronucleation and chromosome pulverization, phenotypes associated with RP-6306 treatment in *CCNE1*-overexpressing cells[17]. Similarly, a combination of RP-6306 and RP-3500 both increased micronucleation (Supplementary Fig. S6F) and chromosome pulverization (Fig. 4E) in *CCNE1*-overexpressing cells with little effect in the wildtype parental cells. We also observed increased γH2AX in OVCAR3

and WU-115 xenografts with the RP-6306/RP-3500 combination compared to either single agent alone, suggesting that DNA damage accumulates in PKMYT1i-ATRi treated tumors (Fig. 4F, G). DNA damage was accompanied by apoptosis in PKMYT1i-ATRi treated *CCNE1*-amp and *CCNE1*-overexpressing cells, as demonstrated by increased Annexin V staining (Fig. 4H, I and Supplementary Fig. S7A) and elevated cleaved PARP, caspase-9 and caspase-3 (Fig. 4J, K). PKMYT1i-ATRi combination treatment-induced apoptosis is dependent on caspases as pan- caspase inhibitor, Z-VAD-FMK, abrogated PKMYT1i-ATRi induced apoptosis (Supplementary Fig. S7B). Importantly, high levels of apoptosis were dependent on *CCNE1* amplification (Fig. 4H and Supplementary Fig. S7A) or overexpression (Fig. 4I and Supplementary Fig. S7A). Together, these results suggest PKMYT1i-ATRi induces lethal amounts of DNA damage in cells with elevated *CCNE1* copy number or cyclin E expression.

Previous studies using ATRi in combination with WEE1i or agents that induce replication stress established that irreversible levels of DNA damage arise from DNA replication defects and exhaustion of the available pool of replication protein A (RPA), leading to replication catastrophe from the conversion of single-stranded DNA (ssDNA) to double-strand breaks (DSBs)[11,32]. We tested if the source of pan-γH2AX in PKMYT1i-ATRi treated *CCNE1*-overexpressing or *CCNE1*-amplified cells originated from replication catastrophe by simultaneously measuring chromatin-bound RPA and γH2AX using QIBC (Supplementary Fig. S8A–D). Cells treated with RP-3500 and hydroxyurea to induce replication stress showed the characteristic replication catastrophe profile with the emergence of cells with pan-RPA preceding those with pan-γH2AX (Supplementary Fig. S8A–D). In contrast, the majority of PKMYT1i and PKMYT1-ATRi treated *CCNE1*-overexpressing FT282 and OVCAR3 cells accumulated pan- γH2AX before the appearance of pan-RPA suggesting that RPA exhaustion is not causing DNA damage (Supplementary Fig. S8A–D). We note there is slight induction of pan-RPA⁺/γH2AX⁻ cells in ATRi-treated cells at later time points, indicating a small contribution of replication catastrophe. We conclude the predominant mechanism of pan-γH2AX induction in PKMYT1i-ATRi treated *CCNE1*-overexpressing or *CCNE1*-amplified cells do not result from replication fork breakage or replication catastrophe.

PKMYT1 inhibition in *CCNE1*-overexpressing or *CCNE1*-amplified cells activates CDK1 in S-phase triggering premature mitotic entry leading to chromosome pulverization and cell death[17]. Considering ATR inhibits cell cycle progression and CDK1 activation during S-phase[33,34], we examined if premature mitotic entry is causing DNA damage in PKMYT1i-ATRi treated cells by measuring the proportion of EdU-positive (EdU⁺) cells marked by histone H3 Ser10 phosphorylation (H3-pS10, Supplementary Fig. S8E). PKMYT1i treatment alone increased premature mitotic entry of *CCNE1*-overexpressing FT282 and OVCAR3 cells based on the emergence of histone H3 Ser10 phosphorylation (H3pS10⁺) in EdU⁺ cells (Fig. 5A). Importantly, the addition of ATRi at doses that have no single agent effect increased the proportion EdU⁺/H3pS10⁺ cells. Increased H3-pS10 expression was also

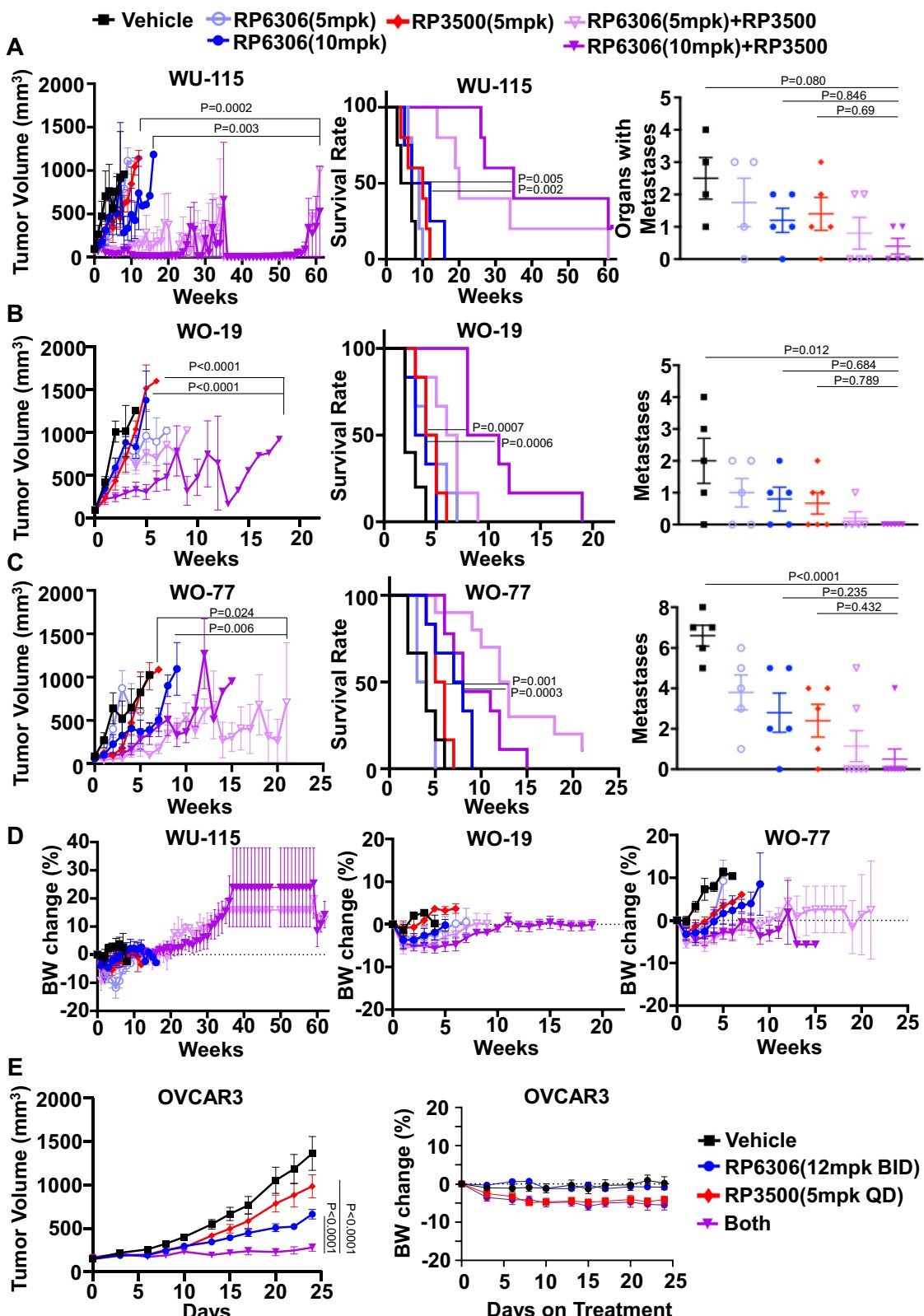

observed in PKMYT1i-ATRi treated *CCNE1*[AMP] (KLE and FUOV1) cells but not in *CCNE1*[GAIN] (OVCAR8) or *CCNE1*[LOW] (WO-20) cells indicating a conserved mechanism-of-action in tumor-derived *CCNE1*-amplified models (Fig. 5B). To further understand the effect of combined PKMYT1-ATRi on mitotic entry we conducted time-lapse microscopy of cells expressing a PCNA chromobody fused to TagRFP[17]. We define a premature mitotic event when there is nuclear envelope breakdown in

a cell with PCNA puncta, a marker for active DNA replication and S-phase. In concordance with the QIBC results, we observed that combined RP-6306 with RP-3500 treatment increased the frequency of premature mitosis in either the first or second S-phase of *CCNE1*-overexpressing cells compared to either single agent alone or the combination in wildtype parental cells (Fig. 5C, supplemental videos 1–6). Finally, we measured DNA replication fork progression with a

**Fig. 3 | Combination of PKMYTi-ATRi synergistically suppress CCNE1 amplified OVCA and EMCA PDXs growth. A**−**C** Tumor volume growth was measured weekly in (**A**) EMCA WU-115 (**B**) OVCA WO-19, and (**C**) OVCA WO-77 *CCNE1* amplified PDX models treated with the indicated drugs. RP-6306 was given oral BID on days 1−5, and RP-3500 was given oral QD on 3 days on / 4 days off schedule until tumor progression (tumor volume >1000 mm³). Survival rate (middle panel) and metastases (right panel) was analyzed at the end of each experiment. Metastases was defined as the number of organs with metastatic disease in each mouse. **D** The toxicity of drugs was revealed by the mice body weights changes. **E** Tumor growth

(left) and body weight change (right) of OVCAR3 xenografts in mice treated with either RP-6306, RP-3500, or both. RP-3500 was given oral QD, and RP-6306 was given oral BID, both given on a 3-day on / 4-day off schedule for 24 days (*n* = 8). Tumor growth and percent body weight change shown is mean ± SEM. Longitudinal tumor growth was analyzed by linear mixed effects modeling with type II ANOVA and pairwise comparisons across groups. Data were analyzed for overall survival using the Mantel-Cox log-rank test. Metastases were compared with one-way ANOVA followed by Tukey's multiple comparisons. The body weight shown is mean ± SEM.

---

combination of RP-6306 and RP-3500 in OVCAR3 cells using DNA fiber assays[35]. OVCAR3 cells were treated with RP-6306, RP-3500, or the combination and sequentially labeled with the nucleotide analogs CldU and IdU. Consistent with an interruption of DNA replication for progression, we observe shorter CIdU+IdU track lengths in OVCAR3 cells treated with combination RP-6306 and RP-3500 compared to monotherapy or untreated controls (Fig. 5D). Taken together, these results suggest that ATR is reducing the CDK1 activation potential of RP-6306 and limiting induction of premature mitosis in *CCNE1*-overexpressing or *CCNE1*-amplified cells.

### RP-3500 cooperates with RP-6306 to activate CDK1 in S-phase

We postulated that cyclin B-CDK1 activation in the S-phase precedes premature mitotic entry in *CCNE1*-overexpressing or *CCNE1*-amplified cells treated with PKMYT1i-ATRi. At the onset of prophase, cyclin B-CDK1 complexes are rapidly activated and imported into the nucleus marked by CDK1 autophosphorylation of cyclin B on Ser126[36,37] (cyclin B-pS126⁺). In PKMYT1i treated *CCNE1*-overexpressing FT282 and OVCAR3 cells, cyclin B-pS126 accumulated in the nucleus of EdU⁺ cells (Fig. 6A and Supplementary Fig. S9A), and the addition of ATRi increased the proportion of EdU⁺/cyclin B-pS126⁺ cells. Only a mild increase in EdU⁺/cyclin B-pS126⁺ cells was observed at later time points in FT282 parental cells, indicating that high CCNE1 expression underpins the robust and premature CDK1 activation by PKMYT1i-ATRi.

To investigate how PKMYT1 and ATR inhibition are cooperating to activate CDK1, we measured levels of the CDK1 inhibitory phosphorylation at Thr14 (CDK1-pT14) in cell lines and tumor xenografts. As expected, PKMYT1i treatment reduced CDK1-pT14 levels in *CCNE1*-overexpressing/*CCNE1*-amplified cell lines (Fig. 6B−D and Supplementary Fig. S9B) and *CCNE1*-amp OVCAR3, WO-77 and WU-115 xenografts (Fig. 6E, F and Supplementary Fig. S9C−E). Remarkably, the addition of ATRi to PKMYT1i facilitated a greater reduction in CDK1-pT14 levels compared to PKMYT1i alone in cell lines (Fig. 6B−D and Supplementary Fig. S9B), suggesting that ATRi is bolstering PKMYT1i-dependent dephosphorylation and activation of CDK1. We also see decreased CDK1-pT14 levels in WO-77 and WU-115 xenografts (Fig. 6E, F and Supplementary Fig. S9D, E). In response to DNA damage, ATR activates CHK1, which halts G2/M progression by catalyzing inhibitory phosphorylation of CDC25B/C phosphatases[38]. CDC25B and CDC25C phosphatases act in a sequential and coordinated manner to activate CDK1 by dephosphorylating CDK1. We reasoned that the DNA damage generated by PKMYT1i treatment leads to ATR-CHK1 activation and CDC25B inhibition, which is blocked by ATRi treatment. We investigated this by monitoring the activating phosphorylation of Ser345 on CHK1 (CHK1-pS345) and inhibitory phosphorylation of Ser151 on CDC25B (CDC25B-pS151). In *CCNE1*-overexpressing FT282 cells both CHK1-pS345 and CDC25B-pS151 levels increased upon PKMYT1i treatment and were partially suppressed by the addition of ATRi (Fig. 6B and Supplementary Fig. S9B). CHK1-pS345 levels were also reduced by ATRi addition to PKMYT1i in *CCNE1*-amp OVCAR3 and KLE cells (Fig. 6C, D). Finally, co-treatment of *CCNE1*-overexpressing FT282 cells with the CDK1 inhibitor RO-3306 significantly reduced PKMYT1i-ATRi-dependant pan-γH2AX induction (Supplementary Fig. S9F). Together, these results suggest that the addition of ATRi to PKMYT1i

permits rapid and robust CDK1 activation by stimulating CDK1 dephosphorylation.

Our current studies identified a strong synergistic interaction between low doses of PKMYT1i-ATRi in *CCNE1*-overexpressing or *CCNE1*-amplified preclinical models. We compared the sensitivity of PKMYT1-ATRi combination in an isogenic panel of RPE1-hTERT *TP53*⁻/⁻ parental, *ATM*⁻/⁻, *BRCA1*⁻/⁻ *and CCNE1*-overexpressing cells (Supplementary Fig. S10A). The concentration of PKMYT1i that attained the highest synergy was lower in *CCNE1*-overexpressing compared to parental, *ATM*⁻/⁻ and *BRCA1*⁻/⁻ cells, indicating the synthetic lethal window of RP-6306 in *CCNE1*-overexpressing cells was exacerbated by the addition of ATRi (Fig. 6G and Supplementary Fig. S9B). For example, combining 24.7 nM PKMYT1i with 12.3 nn ATRi was cytotoxic in *CCNE1*-overexpressing but not in the parental, *ATM*⁻/⁻ or *BRCA1*⁻/⁻ counterparts (Fig. 6G). These results suggest that CCNE1-amplification, rather than ATM or BRCA1/2 inactivation, may associate with increased tumor sensitivity to PKMYT1i-ATRi combinations.

In summary, we propose a model where the synergistic cytotoxicity of PKMYT1i-ATRi originates from the ability of ATR assisting PKMYT1 in keeping CDK1 activity low during S-phase in *CCNE1*-overexpressing cells (Fig. 7). PKMYT1i causes DNA damage that activates ATR-CHK1 and represses CDC25 phosphatases limiting CDK1 activation potential. Combined PKMYT1i-ATRi increases CDC25 phosphatase activity, allowing deeper dephosphorylation of CDK1-pT14, which rapidly drives S-phase cells into mitosis resulting in catastrophic DNA damage and cell death (Fig. 7).

## Discussion

*CCNE1* is commonly amplified in gynecological cancers such as OVCA and EMCA, and effective treatments exploiting this genomic alteration are lacking[5,6,39]. Given that this subset of cancers are typically resistant to standard-of-care platinum chemotherapy and associated with poor overall survival, we sought to address this clinical unmet need by identifying a treatment strategy that targets critical survival pathways for *CCNE1*-driven cancers[5,6]. We previously identified PKMYT1 as a new synthetic lethal target for *CCNE1* amplified cells using a genome-wide CRISPR screen approach[17,28]. PKMYT1 inhibition with RP-6306 is a selective and potent CDK1 activator, leading to mitotic catastrophe, especially in *CCNE1* amplified models[17]. Because the emergence of resistance to monotherapy for oncogene-addicted cancers is essentially universal[40], a combination strategy was investigated. Further, combination strategies that exploit genomic vulnerabilities can permit the utilization of drug concentrations lower than required to be active as monotherapy, thereby decreasing toxicity[41]. Considering *CCNE1* amplification causes replication stress that activates the DNA replication fork stabilizer and regulator of G2/M checkpoint kinase ATR, we sought to further improve anti-tumor efficacy by combining PKMYT1 inhibition (RP-6306) with ATR inhibition (RP-3500).

Here, we demonstrate that combined inhibition of PKMYT1 and ATR is synergistic in *CCNE1* amplified OVCA and EMCA cells compared to non-amplified models and effects are CCNE1 copy number dependent (Fig. 1C−F, Supplementary Fig. S1C, D and Fig. 2C, E−I). We demonstrate a significant increase in anti-tumor activity and overall survival compared to monotherapy alone in *CCNE1* amplified OVCA

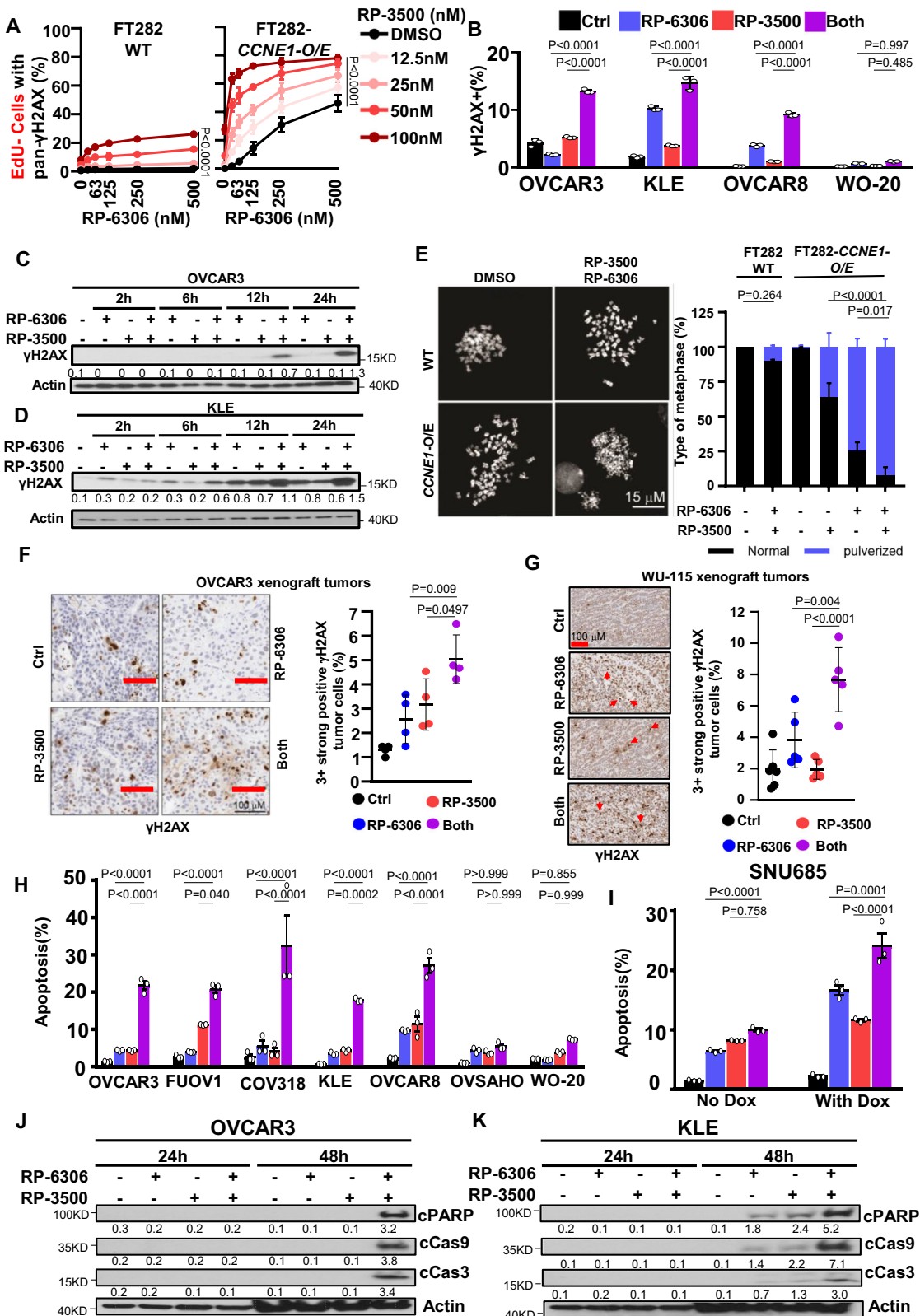

and EMCA PDX models using a low-dosing strategy justifying further evaluation in the clinic (Fig. 3 and Supplementary Fig. S5). Notably, especially in highly metastatic PDX models, a combination of PKMYT1i and ATRi decreased metastasis in OVCA and EMCA PDX models which is clinically meaningful as metastatic disease is what ultimately kills our patients. We observe that a combination of PKMYT1 with ATR inhibition leads to defective DNA replication

(Fig. 5C, D) and induces lethal amounts of DNA damage in cells with elevated *CCNE1* copy number or cyclin E expression as evidence by increased γH2AX (Fig. 4A–D and Supplementary Fig. S6A–E) and Annexin V (Fig. 4H, I) as well as cleaved caspase 3 (Fig. 4J, K and Supplementary Fig. S7A). Notably, the most dramatic increase in DNA damage and cytotoxicity in *CCNE1*- overexpressing/amplified cells is observed at PKMYT1i and ATRi concentrations that have little effect

**Fig. 4 | Dual inhibition of PKMYT1 and ATR induced DNA damage and cell apoptosis in CCNE1 amplified cancers. A** QIBC quantitation of FT282-hTERT p53[R175H] parental (WT, left) and *CCNE1*-overexpressing (*CCNE1*-O/E, right) EdU/pan-γH2AX+ cell in response to the indicated RP-6306/RP-3500 combinations treated for 48 h. *n* = 3; Mean + SD. **B** Detection of γH2AX+ cells by flow cytometry in indicated cells after treated with RP-6306 (250 nM), RP-3500 (50 nM), or a combination of both treated for 24 h. *n* = 3; Mean + SD. **C, D** Whole cell lysates of OVCAR3 (**C**) and KLE (**D**) cells were treated with RP-6306 (250 nM), RP-3500 (50 nM), or both for the indicated times and immunoblotted with γH2AX and Actin antibodies. Actin is loading control. **E** Representative micrographs (left) of metaphase spreads from FT282 parental (WT) and *CCNE1*-overexpressing cells left untreated or following treatment with combination of RP-6306 (125 nM) and RP-3500 (25 nM) for 24 h and quantitation of cells (right) after 24 h treatment with the indicated RP-6306 (125 nM) and RP-3500 (25 nM) conditions with at least 40 metaphases counted per replicates. *n* = 3; Mean + SD. **F** OVCAR3 tumor-bearing mice were administered RP-6306 (5 mg/kg) orally BID, RP-3500 (5 mg/kg) orally QD or a combination of both for 3 days, sacrificed 2 h post last treatment, and tumor tissue was prepared for FFPE. Tumor tissues were stained with γH2AX antibodies (left), and the percentage of γH2AX 3 + strong positive tissue (right) present in the tumor area was quantified by HALO software, *n* = 6,5,5,5;); Mean + SD. Scale bar: 100 μm. **G** WU-115 were administered RP-6306 (10 mg/kg) orally BID, RP-3500 (5 mg/kg) orally QD or a combination of both for 10 days, sacrificed 2 h post last treatment and tumor tissue was prepared for FFPE. Tumor tissues were stained with γH2AX antibodies (left), and the percentage of γH2AX 3 + strong positive tissue (right) present in the tumor area was quantified by HALO software. *n* = 6,5,5,5; Mean + SD. Scale bar: 100 μm. Red arrows illustrate γH2AX 3 + strong positive cells quantified (**H**) Flow cytometry quantification of apoptotic cells with Annexin V and propidium iodide (PI) staining of the indicated cells after treated with drugs RP-6306 (250 nM), RP-3500 (50 nM), or both for 72 hrs. *n* = 3; Mean + SD (**I**) SNU685 CCNE1 inducible cells were treated and detected with apoptotic cells same as (**G**). *n* = 3; Mean + SD. **J, K** Whole cell lysates of OVCAR3 (**J**) and KLE (**K**) cells were treated with RP-6306 (250 nM), RP-3500 (50 nM), or both for the indicated times and immunoblotted with cleaved PARP (cPARP), cleaved caspase 9 (cCas9), cleaved caspase 93 (cCas3), and Actin antibodies. Actin is loading control. Significance determined by two-way ANOVA followed by Tukey's multiple comparisons test for (**A, B, E, H, I**) and one-way ANOVA followed by Tukey's multiple comparisons for OVCAR3 and WU-115 xenograft in (**F, G**).

as a single agent or in immortalized fallopian tube cells (FT282) without *CCNE1* overexpression (Figs. 2 and 4), thus suggesting clinical tolerability.

Other effective combination studies targeting *CCNE1* amplification and or expression have been identified preclinically. We recently showed that CCNE1 amplification is a biomarker of response to the combination WEE1 with ATR inhibition (WEE1i-ATRi)[11]. Mechanistically, the WEE1i-ATRi combination anti-tumor effects differ from the PKMYT1i-ATRi combination. We previously showed that combination WEE1i-ATRi treatment results in irreversible levels of DNA damage that arise from DNA replication defects and exhaustion of the available pool of RPA, ultimately leading to replication catastrophe from conversion of single-strand DNA to double-strand breaks (DSBs)[11,32]. In this study, we demonstrate that the major mechanism of action of PKMYT1i-ATRi does not result from replication fork breakage or replication catastrophe, as evidenced by the accumulation of pan-γH2AX before the appearance of pan-RPA (Supplementary Fig. S8A–D). The differential mechanism of action between PKMYTi-ATRi and WEE1i-ATRi can potentially be attributed to the observations that WEE1i increases origin firing and DNA replication stress via activation of Cyclin E-CDK2 leading to greater reliance on ATR to stabilize and restrict replication fork progression[42,43], whereas PKMYT1i has little effect on CDK2 activation[17]. Monotherapy of both WEE1 and ATR inhibitors are both associated with myelosuppression, suggesting an overlapping toxicity profile[6,7]. Conversely, RP-6306 monotherapy does not lead to substantial myelosuppression[42] suggesting a more favorable toxicity profile for combination with ATR inhibitors (or WEE1). Taken together, this study indicates that CCNE1 amplified OVCA and EMCA cells are specifically vulnerable to CDK1 activation, and strategies to increase CDK1 activation offer an attractive therapeutic avenue for this unmet need in the clinic. Preclinically, other combinations targeting CCNE1 overexpression include CDK2 inhibition (e.g., dinaciclib) with AKT inhibition, CDK2/9 and PIK3CA inhibition, and WEE1 with PKMYT1 inhibition[44–47].

There are several strategies targeting *CCNE1* amplification or CCNE1 overexpressing solid tumors that are being evaluated in the clinic in Phase I/II trials. Targeting WEE1 with ZN-c3, CHK1/2 with LY2606368, and CDK2 with INX-315, BLU-222, INCB123667, or ARTS-021 are all in early phase I/II monotherapy clinical trials (clinical-trials.gov). Thus far, CDK2 inhibitors have largely failed in clinical trials due to insufficient selectivity[10]. Clinical trials of combination regimens targeting CCNE1 are also in development. Given the results of this study showing that *CCNE1* amplification or overexpression represents a strong biomarker for sensitivity to low-dose combination PKMYT1i (RP-6306) with ATRi (RP-3500), this combination has moved forward

into the clinic as a phase 1 dose escalation study in advanced solid tumors with *CCNE1* amplification (NCT04855656; MYTHIC). PKMYT1i, in combination with the chemotherapies gemcitabine and FOLFIRI, is also being explored given preclinical data showing this combination is synergistic and similarly resulted in mitotic catastrophe (MAGNETIC and MINOTAUR: clinical trials.gov)[17]. Chemotherapy combinations with targeted agents such as older generation WEE1i have demonstrated activity but have been overall intolerable because of toxicity[43].

Our data demonstrates that *CCNE1* gene amplification and overexpression are important biomarkers for sensitization to PKMYT1 combined with ATR inhibition. We show that mutation of *BRCA1* or *ATM*, the clinical biomarkers for ATRi or ATRi-PARPi sensitivity, show little to no sensitivity to low doses of PKMYT1i-ATRi. Considering the synthetic lethal relationship between *CCNE1 -BRCA1* and the near mutually exclusivity of *CCNE1* amplification and *BRCA1* mutations in tumors[44], results from our work support the inclusion of tumors with *CCNE1* amplification and exclusion of those with homologous recombination or ATM deficiencies for combination PKMYT1 with ATR inhibition. However, there is a clinical need to determine if *CCNE1* overexpression, either by gene copy number (CN) or protein level better correlates with response to agents targeting this oncogene. The optimal copy number threshold for *CCNE1* amplification and the role of cyclin E protein levels as predictive biomarkers of response is currently being investigated across preclinical and clinical studies. Further, it is possible that other biomarkers exist to predict sensitivity to this drug combination. Low molecular weight Cyclin E1 has been shown to facilitate replication stress tolerance and DNA damage repair, suggesting sensitivity to drugs targeting the ATR/CHK1 pathway[45,46]. We have previously identified both full-length and low molecular weight isoforms in the Cyclin E1 overexpressing cell lines used in this study[11,17]. The scope of our study was limited to full-length Cyclin E1 and does not exclude the possibility that other biomarkers could enhance response[46]. Other mutations have also been identified that may predict sensitivity to PKMYT1i or the combination, such as *FBXW7* (which encodes a substrate adapter for the E3 ligase that targets cyclin E for ubiquitin-dependent proteolysis)[47], or *PPP2R1A* (a serine/threonine phosphatase and tumor suppressor)[17]. Cells evaluated such as KLE (*CCNE1*[AMP]) and SKOV3 (*CCNE1*[LOW]) have a likely pathogenic mutation in *FBXW7* and OVCAR3 (*CCNE1*[AMP]), a *PPP2R1A* mutation which has not been further characterized; such alterations all potentially increasing the sensitivity to PKMYT1i or combination[48].

There are limitations to 2-dimensional cultures, which does not fully recapitulate the complex cell-cell and cell-environment interactions of a 3-dimensional or in vivo systems, which may affect the

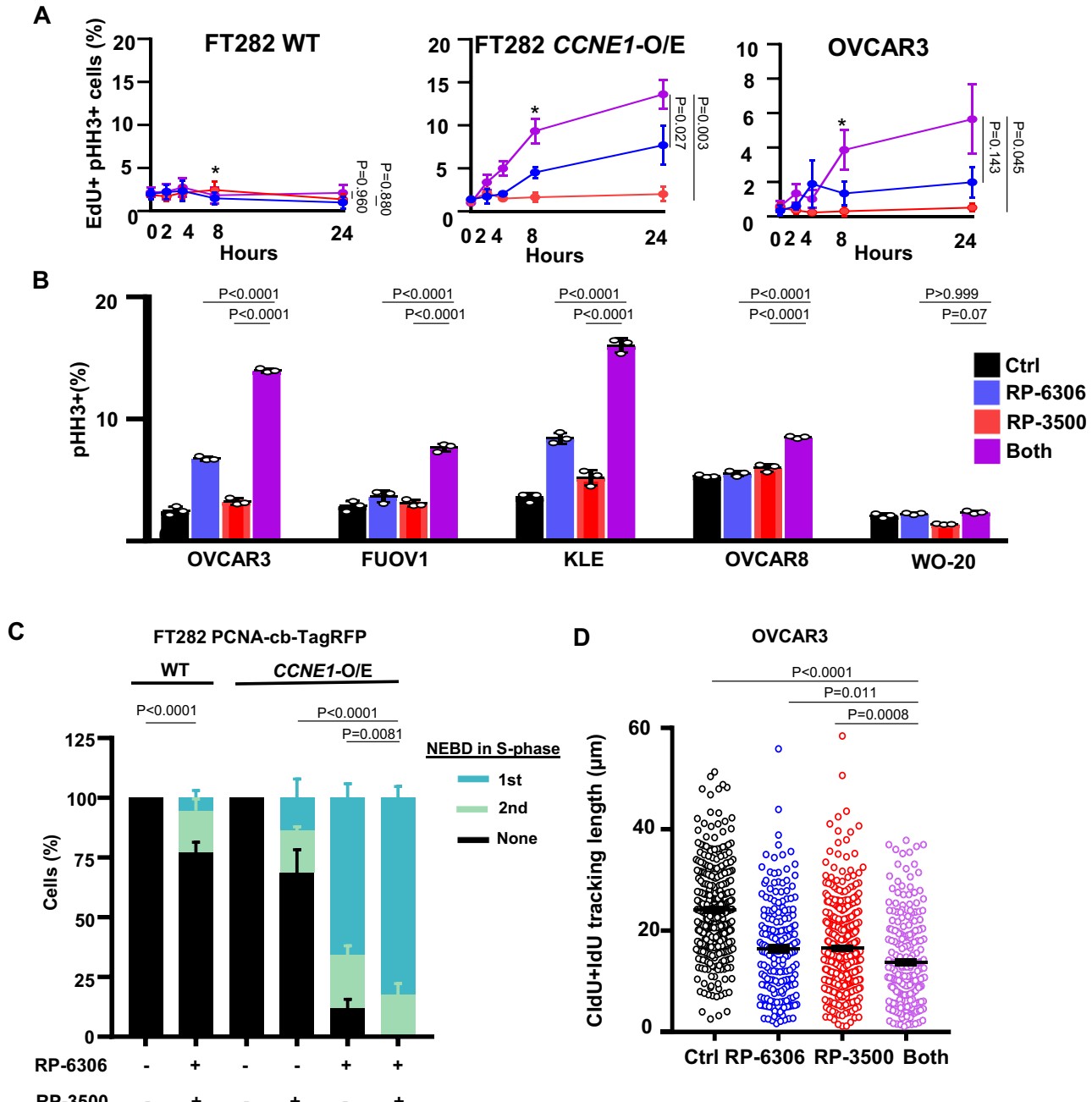

**Fig. 5 | PKMYT1i-ATRi combination resulted double strand DNA in CCNE1 amplified OVCA and EMCA. A** QIBC quantitation of FT282-hTERT $p53^{R175H}$ parental (WT, left) *CCNE1*-overexpressing (*CCNE1*-O/E, middle) and OVCAR3 (right) cells with percent of EdU$^+$/ pHH3$^+$ as a function of time after addition of RP-6306 (250 nM), RP-3500 (100 nM) or combination of both. * *P*-values reveal the comparison of groups at 8 h. **B** Measurement pHH3$^+$ cells in the indicated cells after treated with RP-6306 (250 nM), RP-3500 (50 nM), or combination for 24 h. *n* = 3; Mean + SD. **C** Quantitation of the number of nuclear envelope breakdowns (NEBDs) observed during the 1$^{st}$ or 2$^{nd}$ observed S-phase using time-lapse imaging of FT282-hTERT

$p53^{R175H}$ PCNA-chromobody-TagRFP (WT) and *CCNE1*-overexpressing (*CCNE1*) cells treated with the indicated RP-6306 (125 nM) or RP-3500 (25 nM) for 47 h. *n* = 3; Mean + SD. **D** DNA fiber assay were performed to detect DNA fiber progression. The OVCAR3 cells were treated with RP-6306 (250 nM), RP-3500 (50 nM), or a combination for 1 h, then pulsed with CldU (red) and IdU (green) for 25 min. Significance determined by one-way ANOVA followed by Tukey's multiple comparisons test in (**A**, **D**), and two-way ANOVA followed by Tukey's multiple comparisons test for (**B**, **C**).

efficacy and synergy of drug combinations. Cancer's molecular landscape, heterogeneity, and tumor microenvironment all influence tumorigenesis, metastasis, response to treatment, and emergence of drug resistance[49]. Further, DNA damage repair inhibitors may influence immune system response[50], which is lacking in the models used in this study.

In summary, we identify a potential treatment option for an aggressive subset of OVCA and EMCA patients who have poor

prognosis and limited treatment options. By exploiting oncogene-addicted cell-cycle checkpoints and DNA repair mechanisms with a combination of PKMYT1 with ATR inhibition, normal cells should be spared, allowing lower dosing strategies, thereby limiting toxicity. Translational endpoints in ongoing and future clinical trials with this drug combination and additional preclinical studies are crucial to define the optimal *CCNE1* CN (or protein) level to predict sensitivity to this drug combination.

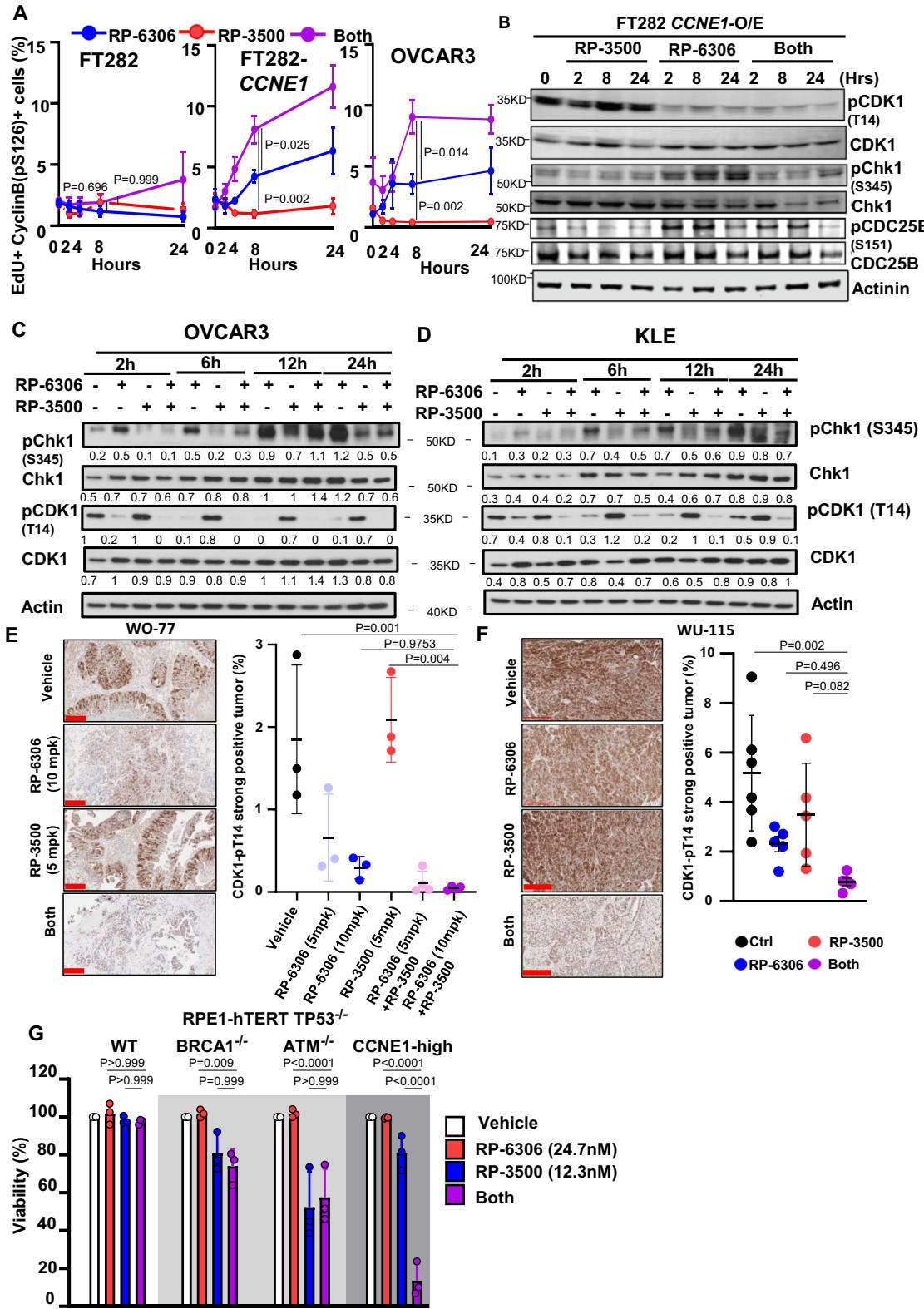

## Methods

### PDX Studies and Ethics statement

NSG mice (NOD/SCID IL2Rγ − /−) were purchased from the Stem Cell and Xenograft Core (SCXC) at the University of Pennsylvania (UPENN, Philadelphia, PA). All mice experiments were performed in adherence to the policies of the NIH Guide for the Care and Use of Laboratory Animals and approved by the Institutional Animal Care and Use Committee (IACUC). Patients were consented, and tumors were obtained from ovarian cancer debulking surgeries conducted at the Hospital of the UPENN and Pennsylvania Hospital (IRB# 702679).

Orthotopic PDX models were generated by surgically engrafting of patient tumor chunks (3–4 pieces, 2 mm3 each) to the ovary/oviduct of five eight-week-old female mice as previously described[11,51].

**Fig. 6 | PKMYT1i-ATRi leads to premature mitosis in CCNE1 amplified OVCA and EMCA. A** QIBC quantitation of FT282-hTERT *p53*[R175H] parental (WT, left) *CCNE1*-overexpressing (*CCNE1*-O/E, middle) and OVCAR3 (right) cells with percent of EdU[+]/cyclin B-pS126[+] as a function of time after addition of RP-6306 (250 nM), RP-3500 (100 nM) or combination of both. **B** Whole cell lysates of FT282-hTERT *p53*[R175H] CCNE1-overexpressing (*CCNE1*-O/E) cells treated with RP-3500 (100 nM), RP-6306 (250 nM) or both for the indicated times were immunoblotted with CDK1, CDK1-pT14, CHK1, CHK1-pS345, CDC25B, CDC25B-pS151 and Actinin specific antibodies Actinin is used as loading control. **C, D** Whole cell lysates of KLE (C) and OVCAR3 (D) cells treated with RP-6306 (250 nM), RP-3500 (50 nM), or both for the indicated times were immunoblotted with CDK1, CDK1-pT14, CHK1, CHK1-pS345 and Actin specific antibodies. **E, F** Tumor tissue from WO-77 (**E**) tumor-bearing mice from

Fig. 3C at end of treatment or WU-115 (**F**) tumor-bearing mice administered RP-6306 (10 mg/kg) orally BID, RP-3500 (5 mg/kg) orally QD or combination of both for 10 days and sacrificed 2 h post last treatment was prepared for FFPE Tumor tissues were stained with CDK1-pT14 antibodies (left) and the percentage of CDK1-pT14 strong-positive tissue (right) present in the tumor area was quantified by HALO software. *n* = 3,3,3,3,4,4 (**E**), *n* = 6,5,5,5 (**F**) Mean ± SD. Scale bar: 200 μm. **G** Growth inhibition relative to DMSO control of RPE1-hTERT *TP53*[-/-] parental, *BRCA1*[-/-], *ATM*[-/-] and CCNE1-overexpressing (*CCNE1-2A-GFP*) cells after treatment with the indicated dose of RP-6306, RP-3500 or the combination of both. *n* = 3; Mean + SD. Significance determined by one-way ANOVA followed by Tukey's multiple comparisons test in for (**A, E, G**), and Students' *t* test in (**F**).

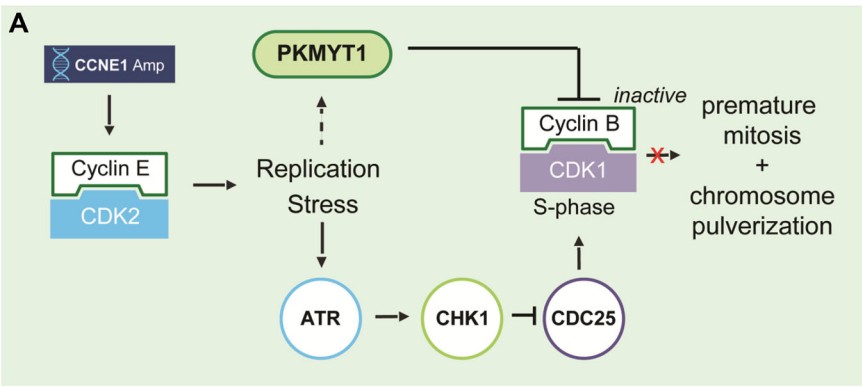

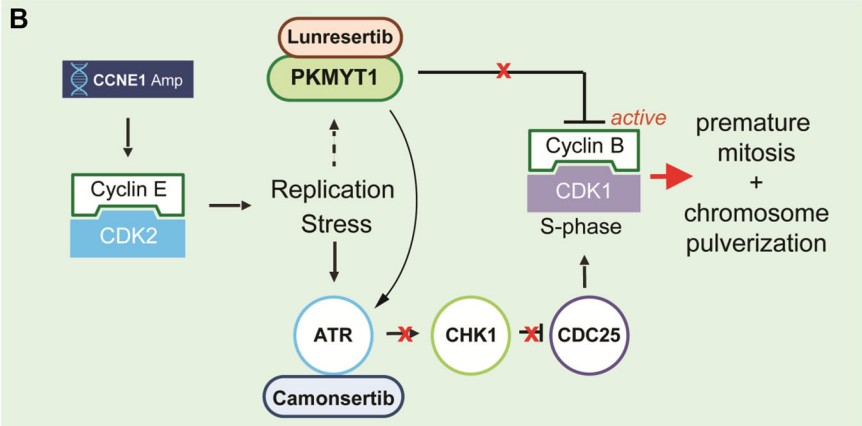

**Fig. 7 | Schematic model of targeting CCNE1 amplified cancers with dual inhibition of PKMYT1 and ATR. A, B** Model of synergy between PKMYT1 and ATR inhibition in *CCNE1*-amplified or overexpressing cells. **A** CCNE1 amplification or overexpression in cells causes replication stress and S-phase elongation. To delay induction of mitosis until DNA replication is complete, CDK1 activity is inhibited by

increased PKMYT1 inhibitory phosphorylation on CDK1-Thr-14 and decreased CDC25 phosphatase activity via ATR-CHK1 signaling. **B** Inhibition of PKMYT1 (lunresertib) reduces CDK1 The14 phosphorylation and inhibition of ATR (camonsertib) increases CDC25 activity resulting in rapid and robust S-phase CDK1 activation and premature mitosis with synergistic induction of DNA damage and cell death.

The harvested PDX tumors were either retransplanted to NSG mice for further expansion or cryopreserved for future use.

We used in this study two high-grade serous ovarian cancer (HGSOC, WO-19, WO-77) PDX models and one endometrial cancer (EMCA, WU-115) PDX model. For preclinical trials, cryopreserved PDX tumor tissue was thawed and transplanted. After tumors were palpable (~ 3-4 mm), tumor volume was measured weekly by ultrasound (SonoSite Edge II Ultrasound System) by a trained sonographer. The tumor volume criteria for randomization to treatment arms was 50-100 mm3. Animals were randomized in a blinded manner into 6 treatment groups: vehicle (0.5% methyl cellulose); RP6306 (10mgkg, BID/day 1–5 weekly), RP6306 (5 mg/kg, BID/day 1–5 weekly), RP3500 (5 mg/kg, QD/day 1–3 weekly), combination RP6306 (10 mg/kg, BID/

day 1–5) + RP3500 (5 mg/kg, QD/day 1–3), combination RP6306 (5 mg/kg, BID/day 1–5) + RP3500 (5 mg/kg, QD/day 1–3). Drugs were dosed by oral gavage. In all the models, the percentage change in body weight during treatment was used as a marker for toxicity and dose level adjustments. Significant treatment toxicity was defined as a 15% drop in body weight, and the mice require treatment reduction at a 25% dose and supplements supportive. For mice with a 20% drop in body weight, treatment was stopped and supportive measures (i.e., food supplement and subcutaneous fluid) were provided. The body weights and condition scores of mice were monitored and recorded weekly. Once improved, treatment was restarted with a 25% dose reduction. If body weight was not regained after one week, the animal was sacrificed in accordance with the Institutional Animal Care and Use Committee

(IACUC) protocols. Trial endpoints were defined as tumor volume > 1000 mm3 for all orthotopic PDX models. The humane endpoints for all mouse experiments were never exceeded. At the end of each experiment, mice were also assessed for the presence of ascites (0 no ascites, 1 small ascites, 2 medium ascites, 3 large ascites) and metastases at the time of necropsy. In the WU-115 model, two mice in the RP-6306 (10 mg/kg) + RP-3500 group were monitored without detectable tumors starting at week 38. Their tumors were palpated weekly, but not body weight not measured, which leads to unchanged body weight in from week 38 to week 56 Fig. 3A.

Targeted DNA sequencing and sequencing analysis, variant calling, and copy Number Profiling for PDX tumors was previously reported[11,52].

## Cell lines and primary cells

OVCAR3, FUOV1, and KLE cell lines were purchased from ATCC (Manassas, Virginia); FUOV1 was obtained from Leibniz Institute DSMZ; OVCAR8 was obtained from NCI-DTP; Kuramochi, OVSAHO and OVKATE obtained from the Japanese Collection of Research Bioresources Cell Bank (JCRB). SNU685 from AcceGen Biotech (Fairview, NJ). $CCNE1^{Amp}$ lines (copy number [CN] > 5) were: OVCAR3, FUOV1, COV318, KLE; CCNE1Gain (CN 2-5): OVCAR8, OVSAHO, Kuramochi; CCNE1 copy neutral (CCNE1Low): OVKATE, WO-20, SNU685. Ovarian cancer cell lines included: OVCAR3, FUOV1, COV318, OVCAR8, OVSAHO, WO-20, OVKATE. Endometrial cancer cell lines included: KLE SNU685. OVCAR3, OVCAR8, Kuramochi, OVSAHO, OVKATE, and SNU685 cells were maintained in RPMI 1640 media with 10% fetal bovine Serum (FBS; Thermo Fisher) and 1% penicillin/ streptomycin (P/S; Thermo Fisher). FUOV1 and KLE cells were cultured in Dulbecco's Modified Eagle's Medium (DMEM)/F12 media with 10% FBS and 1% P/S. RPE1-hTERT $p53^{-/-}$ Cas9, RPE1-hTERT $p53^{-/-}$ Cas9 $BRCA1^{-/-53}$, RPE1-hTERT $p53^{-/-}$ Cas9 $ATM^{-/-29}$ and RPE1-hTERT $p53^{-/-}$ Cas9 CCNE1 overexpressing cells[17] were grown in DMEM (Life technologies # 11965-092) with 10% FBS (Wisent #080150) and 1% Pen/ Strep (Wisent #450-201-EL). FT282-hTERT $p53R^{175H}$ WT (empty vector) and CCNE1 overexpressing cell lines were obtained from Ronny Drapkin[13] and cultured in DMEM: F-12(1:1) (Life technologies # 11330-032) with 5% FBS, 1% UltroserG (Pall Life Sciences #15950-017) and 1% Pen/Strep.

The WO-20 primary ovarian cancer tumor cultures were generated in Simpkins laboratory as previous[11]. WO-20 CCNE1 inducible and SNU685 CCNE1 inducible cells were established by lentivirus stable infection[11]. Cell lines were authenticated by short tandem repeat (STR) analysis at the Oncogenomics Core at Wistar Institute and confirmed mycoplasma negative by end-point PCR at the Cell Center Service at the University of Pennsylvania. For established cell lines, CCNE1 copy number data from the Cancer Cell Line Encyclopedia (CCLE) was used. These data are available online, at https://depmap.org. An absolute copy number of ≥6 was considered CCNE1 amplified, CN >2 and <6 copy gain, and ≤2 copy neutral / low.

## Cell line-derived xenografts

Animals were housed, and experiments were performed at Repare Therapeutics (Admare Bioinnovations Montreal site, St-Laurent, Canada), which is a CCAC (Canadian Council on Animal Care) accredited vivarium. Studies were conducted under a protocol approved by the Admare Animal Care Committee (AACC). Mice were inspected upon arrival, and group-housed (3–5 per cage) in individual HEPA ventilated autoclaved cages (Blue Line, Techniplast, Buguggiate, Italy) in a temperature-controlled environment (22 ± 1.5 °C, 30–80 % relative humidity, 12 h light/dark). Animals were provided with autoclaved corncob bedding, irradiated food (Harlan Teklad, Montreal, Canada), and filtered, autoclaved water ad libitum. They were also provided with nesting material and a plastic shelter as enrichment. Fresh bedding, nesting material, and water was replenished/replaced on a weekly

basis. Mice were acclimatized in the animal facility for at least 5 days prior to use and were identified with indelible ink. Experiments were performed during the light phase of the cycle.

OVCAR3 cells were implanted at $5 \times 10^6$ cells per mouse into the right flanks of female SCID-beige mice respectively (5-7 weeks old; Charles River), in 1:1 Matrigel:media (ECM gel Sigma, cat# 1270; media Corning RPMI 1640 cat #10-41-CM). When tumors reached the average target size of ~150 mm$^3$ (between ~100 and 200 mm$^3$), (n = 8) mice were randomized to treatment groups according to tumor volume and body weight using the "stratified" method in Studylogv4.4 software, and treatment with lunresertib and camonsertib was initiated. Lunresertib was formulated in 0.5% methylcellulose and orally administered twice daily (BID, 0-8 h) for cycles of 3 days on/4 days off, for 28 days (4 cycles). Camonsertib was formulated in 0.5% methylcellulose and 0.02% SLS (pH 6.00) and orally administered once daily (QD) for cycles of 3 days on/4 days off, for 28 days (4 cycles). Statistical significance relative to vehicle control and other test groups was established by one-way Brown Forsyth and Welch ANOVA tests followed by unpaired t with Welch's correction, with individual variances computed for each comparison for multiple groups and unpaired t-test for two group comparisons (GraphPad Prism v9.0).

## Blood and tumor tissue collection

Under isoflurane anesthesia, whole blood was collected by cardiac puncture and transferred to tubes containing 0.1 M citric acid (3:1 citric acid:blood) and stored at − 20 °C for LC-MS/MS analysis. Tumors were removed from mice flanks and cleared of surrounding mouse stroma. Tumor pieces between 50 mg and 100 mg were collected in a pre-weighed pre-filled bead mill tube (Fisher Scientific, Cat# 15-340-154) and then flash-frozen in liquid nitrogen. Other tumor fragments from vehicle- and compound-treated mice were placed in 10% neutral buffered formalin (NBF) within 2-3 min of surgical excision, fixed in NBF for 24 hours at room temperature, and embedded in paraffin.

## RP-6306 and RP-3500 Quantitation by LC-MS-MS

The extraction of whole blood samples was performed by protein precipitation using four volumes of acetonitrile. The sample extracts were analyzed using a Transcend LX2 / Ultimate 3000 liquid chromatography system coupled to a Thermo Altis triple quadrupole electrospray mass spectrometer (Thermo Fisher Scientific) operated in positive mode. Separations were performed using a $2 \times 50$ mm, 2.8 μm Pursuit XRS C8 HPLC column (Agilent). A reversed-phase linear gradient of water + 0.1% formic acid and 1:1 acetonitrile:MeOH was used to elute RP-6306, RP-3500, and the internal standards. Samples were quantified against a 12-point linear standard curve and 5 levels of quality control samples. Whole blood concentrations of RP-6306 and RP-3500 were converted to free unbound plasma concentrations using an in vitro derived blood / plasma ratio = 1.2 and fraction unbound ($f_u$) plasma = 0.185 for RP-6306 and blood / plasma ratio = 0.613 and fraction unbound (fu) plasma = 0.00665 for RP-3500 from the CD-1 mouse strain.

## Immunohistochemistry

Histology in Figs. 4F, 6E, and Supplementary Fig. S9C was performed by HistoWiz Inc. Briefly, the formalin-fixed tissues were dehydrated through a 20, 80, 95 and 100 % ethanol series, cleaned in Histoclear, embedded in paraffin then sectioned at 4 μm. Immunohistochemistry for γH2AX and CDK1pT14 were performed on a Bond Rx autostainer (Leica Biosystems) with heat antigen retrieval. Bond polymer refine detection (Leica Biosystems) was used according to the manufacturer's protocol. After staining, sections were dehydrated and film coverslipped using a TissueTek-Prisma and Coverslipper (Sakura). Whole slide scanning (40x) was performed on an Aperio AT2 (Leica Biosystems). Image quantification

analysis was performed using HALO software (Indica Labs). The percentage of tumor cells staining at each of the following four levels was recorded: 0 (no staining), 1 + (weak staining), 2 + (moderate staining,) and 3 + (strong staining). Thresholds for staining intensity were optimized for each xenograft model. Histology in Supplementary Fig. S3B was performed by Mosaic Laboratories (A CellCarta Company). Immunohistochemistry for Cyclin E1 (rabbit clone EP126) was performed in according to Mosaic's standard operating procedures. This assay was designed and validated to be a laboratory-developed test. After heat-induced epitope retrieval, staining was performed on a Bond-RX autostainer (Leica Biosystems) and visualized with DAB chromogen. Slides were then removed from the instrument, dehydrated, cleared and cover-slipped. Stained slides were evaluated by a board-certified pathologist on a semi-quantitative scale, and the percentage of tumor cells staining at each of the following four levels was recorded: 0 (no staining), 1 + (weak staining), 2 + (moderate staining), and 3 + (strong staining). H-Score was calculated based on the summation of the product of the percent of cells stained at each staining intensity using the following equation: (3 x % cells staining at 3 +) + (2 x % cells staining at 2 +) + (1 x % cells staining at 1 +).

## Establishment and characterization of primary ovarian cancer organoids

Tumor tissue samples were set on a sterile petri dish, and necrotic tissue were removed. The tumor was dissected to a 5 mm square under sterile conditions and washed with HBSS. Cleaned tissues were placed in a new petri dish and then minced. The minced tissues were mixed with enzymatic digestion buffer containing HBSS, collagenase 4 (1 mg/ml), and Rock Inhibitor (Y-27632). The mixture was placed in a 50 mL tube in a water bath at 37 °C for 15 min. The mixture was collected and dripped through a cell strainer on a new 50 mL tube to remove any residual tissue. The suspension was centrifuged at $300 \times g$ for 5 min at room temperature, the supernatant was removed. In case of a visible red pellet, erythrocytes were lysed in RBC Lysis buffer for 5 min at room temperature, followed by two wash steps with 10 mL of HBSS and centrifugation at $300 \times g$ for 5 min. The cell pellet was suspended in Matrigel, and 50 μL drops of matrix cell suspension were allowed to solidify on a pre-warmed 6-well plate at 37 °C for 15 min. On stabilization of the Matrigel, we added the organoid medium cocktail[54]. The culture media is Advanced DMEM/F12 (Thermo Fisher Scientific, Cat#12634010), containing 2 mM Glutamax (Thermo Fisher Scientific, Cat# 35050061), 10 mM HEPES(Sigma-Aldrich, Cat# H0887-100ML), 100unit Pen Strep (Gibco, Cat# 15140-122), 100 ng/ml Noggin (PeproTech, Cat#120-10C-100ug),100 ng/ml R-Spondin-1 (Pepro-Tech, Cat# 120-38-100ug), 1X B27(Thermo Scientific, Cat#17504001), 1.25mM N-Ace-L-Cys (Sigma, Cat#A9165-5G), 100 ug/ml Primocin (Invivogen, Cat# ant-pm-1), 10 mM Nicotinamids (Sigma, Cat#N0636-500G), 500 nM A83-01 (Tocris, Cat#2939), 10 ng/ml FGF10 (Pepro-Tech, Cat#100-26-50ug), 10 ng/ml FGF2 (PeproTech, Cat#100-18B), 10uM SB202190 (Sigma-Aldrich, Cat#S7076-5MG),1uM PGE2(Tocris, Cat#2296-10 mg), and 50 ng/ml EGF(PeproTech, Cat#AF-100-15-500ug). The medium was changed every 3–4 days, and the organoids were passaged at a 1:2–3 dilution every 2–4 weeks. For passaging, organoids were mechanically and enzymatically dissociated into small clusters. Matrigel-embedded organoids were suspended in Cell Recovery Solution (Corning, 500 μL/well). The organoid suspension was occasionally mixed with gentle pipetting for 30 min on ice to completely solubilize the Matrigel. The tube was then placed on ice to precipitate the organoids. The supernatant was removed, and organoids were washed with 1 ml cold PBS. The organoid was suspended in Matrigel and plated on 6-well plate.

After established, organoids were processed for paraffin sectioning using standard protocols for characterization. Matrigel-embedded organoids were suspended in Cell Recovery Solution (Corning, 500 μL/well). The organoid suspension was occasionally mixed with gentle pipetting for 30 min on ice to completely solubilize the Matrigel. The tube was then placed on ice to precipitate the organoids. The supernatant was removed, and organoids were washed with cold PBS. Organoids were fixed with 4% paraformaldehyde (PFA) for 20 min at room temperature and solidified using histogel before embedding in paraffin. 5 μm sections were stained with hematoxylin−eosin (H&E) and Antibodies (p53, PAX8, CK7).

### Cell viability of organoids (Cell Counting Kit-8, CCK8 assay)

WO-58, WO-19, and WO-77 organoids were established and passaged as described above. To detect the drugs' effect on organoids, the organoids were mechanically and enzymatically dissociated into small clusters, and resuspended in 500 μl organoid culture media in an EP tube (See above). 500 μl Matrigel was added into the EP tube and mix. Organoids in Matrigel were seeded in 96 well plates at 45 μl per well. After solidifying at 37 degrees, 50 μl organoid culture media were added to each well. The organoids are cultured for 3 days and then treated with RP-6306 (250 nM), RP-3500 (50 nM), or both for 10 days and measured with CCK8 assay. The cells were incubated with CCK8 solution for 12 hrs and then measured with absorbance at 450 nm.

### In vitro cell viability assay (MTT assay)

Cells were seeded into 96-well plates at 5000 cells/well. Cells were treated with control (DMSO), RP-6306, RP-3500, or a combination at indicated concentrations for 5 days. The drugs were clinical grade and obtained from Repare Therapeutics. At the end of the treatment period, an MTT colorimetric assay was performed to detect the cell viability. Cells were incubated with 10 mL of MTT at 5 mg/ml (Sigma Chemical Co., St Louis, MO) for 4 h at 37 degrees. The supernatant was removed, and 100 mL DMSO (Fisher Scientific, Hampton, NH) was used to dissolve the MTT formazan. Absorbance was measured in a microplate reader at a wavelength of 570 nm. Relative cell viability was calculated, with the non-treatment group as a control.

### Colony formation assay

For colony formation assay, cells were plated onto 24-well at 5000 cells/well and cultured overnight in triplicate. They were then treated with DMSO vehicle, RP-6306, RP-3500, or combination as indicated every 3 days for a total of 10 days. Cells were then fixed and stained with 0.1% Crystal violet in 20% methanol solution. The plates were washed, air-dried, scanned, and quantified in ImageJ (National Institutes of Health, Bethesda, MD).

### Western blotting assay

Cells were treated and collected at the indicated time, then washed and incubated with Laemmli Sample Buffer (4% SDS, 20% Glycerol, 0.12 M Tris-HCl at pH 6.8 in distilled water) containing a protease and phosphatase inhibitor cocktail (EMD Millipore, Billerica, MA). After measured protein concentration with BCA kit (BioRad, Hercules, CA), whole cell lysates (15 mg) were separated on reducing 4−15% SDS-PAGE gels, electrotransferred to PVDF membrane (Bio-Rad, Hercules, CA), blocked with 5% BSA (ThermoFisher) in 1x Tris-buffered saline (ThermoFisher) with 0.1% Tween20 (ThermoFisher) (1x TBST), and immunoblotted with respective primary antibodies including anti-pCHK1(S345) (1:1000, Cell Signaling Technology, Danvers, MA, cat.#2348), anti-CHK1 (1:1000, Cell Signaling Technology, cat.#2360), anti-CHK1 (Santa Cruz G-4 sc-8408, 1:500), anti-CHK1-phosphoS345 (Bethyl, cat#2348, 1:500), anti-Alpha Actinin (Millipore Sigma 05-384, 1:10000), anti-CDC25B (Thermofisher OTI6H9 TA8-12352, 1:500), anti-CDC25B-phosphoS151 (Thermofisher PA5-104568, 1:500), anti-pCDK1(T14) (1:1000, Abcam, Cambridge, UK, cat.# ab58509), anti-CDK1(1:1000, Cell Signaling Technology, cat.#9116), anti-

γH2AX(1:2000, Cell Signaling Technology, cat.#9178), anti-cleaved caspase 3(1:500, Cell Signaling Technology, cat.#9664), Cyclin E1(1:2000, Cell Signaling Technology, cat.#4129), anti-Actin(1:50000, Cell Signaling Technology, cat.#3700), anti-ATM (Cell Signaling Technology,cat#2873, 1:500,), anti-BRCA1 (Dan Durocher, University of Toronto, 1:500). After that, membranes were washed and blotted with species-appropriate horseradish peroxidase conjugated anti-rabbit (1:3000, catalog 7074, Cell Signaling Tech), anti-mouse (1:3000, catalog 7076, Cell Signaling Tech) secondary antibodies or anti-mouse Irdye 800CW (LiCOR 926-32210, 1:10000), anti-mouse Irdye 680CW (LiCOR 926-68072, 1:10000), anti-rabbit Irdye 800CW (LiCOR 925-32213, 1:10000) and anti-rabbit Irdye 680CW (LiCOR 926-68073, 1:10000) in 5% BSA in 1x TBST for 1 h, followed by chemiluminescent substrate (Thermo Scientific, Rockford, IL) incubation and film development. Actin or Actinin was used as loading control for the whole cell.

### Flow cytometry detection of intracellular proteins
Cells were seeded in triplicate and then treated with RP-6306, RP-3500, or combination for indicated time. Cells were then trypsinized, fixed, washed and incubated with blocking buffer. Cells were then stained with the following primary antibodies diluted in blocking buffer at 1:300: gH2AX (Cell Signaling Technology, cat# 9718), pRPA32 (S33, Bethyl Laboratories, cat#A300-246A) or phospho-histone H3 (Ser10, Cell Signaling Technology, cat# 53348). The cells were washed, and incubated with secondary antibody goat anti-Rabbit IgG (H + L), Alexa Fluor 647 (ThermoFisher Scientific) for 30 min. The cells were then incubated with 50 mg/mL propidium iodide (Sigma-Aldrich) and subjected to flow cytometry acquisition on BD LSRII (BD Biosciences) and data analysis with FlowJo (Tree Star, Inc., Ashland, OR).

### siRNA transfection
To evaluate the off target effect of PKMYT1 and ATR inhibitors, OVCAR3 cells were transfected with 10 nM PKMYT1 siRNA (Thermo Fisher Scientific, Assay ID: s194984) and/or 10 nM ATR siRNA (Thermo Fisher Scientific, Assay ID 82, AM51331) with Lipofectamine RNAiMAX reagent (Thermo Fisher Scientific) following the transfection protocol. The cell viability and protein expression were detected 48 h and 24 hr post-transfection respectively.

### Apoptosis analysis
Cells were plated, incubated overnight, and treated with DMSO vehicle, 250 nM RP-6306, 50 nM RP-3500, or a combination for 72 h. For low-dosage combination studies, the cells were treated with DMSO, 31.3 nM RP-6306, 6.25 nM RP-3500, or combination for 5 days or 7 days. For detecting caspase-dependent apoptosis, the cells were pretreated with 20 μM Z-VAD-FMK (HY-16658, Medchem Express) for 1 h, and then treated with DMSO or combination RP-6306 (250 nM) + RP-3500 (50 nM) for 72 h. Apoptosis assay was performed with eBioscience Annexin V Apoptosis Detection Kit APC (Invitrogen, 88-8007-74), according to the manufacturer's instructions. Annexin V-APC and propidium iodide labeled cells were detected by BD Accuri C6 Cytometer (BD Biosciences, San Jose, CA). The acquired data was analyzed with FlowJo (Tree Star, Inc., Ashland, OR).

### High content imaging and quantitative image-based cytometry (QIBC)
High-throughput analysis of nuclear γ-H2AX, Histone H3-phosphoS10, and Cyclin B1-phosphoS126 cells was done as preciously described[17]. Briefly, cells were seeded in 96-well plates (3000 cells/well for FT282-hTERT $p53^{R175H}$) and cultured for up to 24–48 h depending on the experiment. Prior to harvesting, cells were pulsed with 20 μM EdU (5-ethynyl-2-deoxyuridine, Life Technologies #A10044) for 30 min followed by the addition of paraformaldehyde (PFA) in PBS to a final concentration of 4% and incubated for 15 min at room temperature (RT). Cells were then rinsed with PBS and permeabilized using 0.3% Triton X-100/ PBS for 30 min. For chromatin-bound γH2AX and RPA measurements, cells were pre-extracted for 15 min on ice with CSK buffer (300 mM sucrose, 100 mM NaCl, 3 mM $MgCl_2$, 10 mM PIPES pH 7.0, 0.5% v/v Triton-X 100) before PFA fixation. Cells were rinsed with PBS and incubated with EdU staining buffer (150 mM Tris-Cl pH 8.8, 1 mM $CuSO_4$, 100 mM ascorbic acid, and 10 μM AlexaFluor 488 azide (Life Technologies, #A20012) for 30 min. Cells were washed with PBS and incubated in blocking buffer (10% goat serum (Sigma #G6767), 0.5% NP-40 (Sigma-Aldrich, #I3021), 5% w/v Saponin (Sigma-Aldrich, #84510), diluted in PBS) for 30 min. Fresh blocking buffer containing primary antibodies was added for 2 h. Primary antibodies including histone H2A.X (phospho-S139, Millipore Sigma #05-636, 1:500 IF), RPA32 (Abcam ab2175, 1:500 IF), Histone H3-phosphoS10 (Cell Signaling Technology #9706, 1:500 IF), Cyclin B1-phosphoS126 (Abcam ab55184, 1:500 IF). Cells were rinsed three times with PBS and then blocking buffer, with secondary antibodies including AlexaFluor488 goat anti-mouse IgG (Thermo Fisher Scientific A11029, 1:1000), AlexaFluor647 goat anti-rabbit IgG (Thermo Fisher Scientific A21244, 1:1000) and AlexaFluor555 goat anti-mouse IgG (Thermo Fisher Scientific A28180, 1:1000). Then 0.4 μg/mL DAPI (4,6-diamidino-2-phenylindole, Sigma-Aldrich, #D9542) was added for 1 h. After rinsing with PBS, immunocomplexes were fixed again using 4% PFA/PBS for 5 min. Cells were rinsed with PBS, wells were filled with 200 μl PBS, and images were acquired at the Network Biology Collaborative Center (LTRI) on an InCell Analyzer 6000 automated microscope (GE Life Sciences) with a 20X objective. Image analysis was performed using Cellprofiler 3.1.9[55] and RStudio v1.2.5019 in a similar manner as previously described[17].

### DNA fiber assay
OVCAR3 cells were treated with DMSO, 250 nM RP-6306, 50 nM RP-3500 or combination for 1 h, pulse-labeled with 300 μM 5-chloro-2′-deoxyuridine (CldU; cat. # C6891, Sigma-Aldrich, St. Louis, MO) followed by 100 μM 5-iodo-2′-deoxyuridine (IdU; cat. # I7125, Sigma-Aldrich, St. Louis, MO) for 25 min each treatment, in the presence of drug. Aspirate the media and wash with cold PBS twice. The cells are trypsinized and resuspended to $1*10^6$/ml in cold PBS. 2 μl cells were put on one edge of the silane-coated slides (5070, Newcomer Supply). The cells were processed as previously[35]. The cells were lyzed with lysis buffer (200 mM Tris–HCl pH 7.4, 50 mM EDTA, and 0.5% SDS), and stretch alongside the slide slowly. The cells are fixed with Methanol:Acetic acid at 3:1 for 10 min after air-drying. The slide immersed in 2.5 M HCl for 1 hr at room temperature and neutralized with 400 mM Tris-HCl pH 7.4 for 10 mins. The slides were blocked with blocking buffer (5% BSA + 10% Goat serum), and then stained with rat anti-CldU antibody (1:200, ab6326, Abcam) overnight at 4 degree, AlexaFluor 647-conjugated anti-rat IgG secondary antibody (1:100, A-21247, ThermoFisher Scientific) for 1 h at room temperature. The slides were further incubated with mouse anti-IdU antibody (1:40, 347580, BD Pharmigen) for 1 hour and AlexaFluor 488-conjugated anti-mouse IgG secondary antibody (1:100, A-11001, ThermoFisher Scientific) for 1 hour. After staining, the slides were mounted with Prolong Gold antifade mountant (P36930, Thermo Fisher Scientific). The labeled fibers were imaged with a Nikon Eclipse 80i microscope. The fiber length were quantified with ImageJ with at least 150 fibers in each group.

### Time-Lapse microscopy
cb-PCNA-TagRFP expressing cells were maintained at 37 °C and 5% CO2 while line scanning confocal microscopy was performed using the Nikon Biopipeline live high-content system equipped with an NA 0.45 20x ELWD objective (Nikon) and a Nikon A1 LFOV imaging system.

A single field was acquired every 10 min over 48 h with a single z-stack (1.244 µm/pixel).

## Cytogenetic analyses

$2 \times 10^6$ FT282-hTERT $p53^{R175H}$ cells were seeded in 10-cm dishes. 24 h later RP-6306 (125 nM), RP-3500 (25 nM), or combination of both was added. 22 h later, 100 ng/mL KaryoMAX colcemid (Thermo Fisher Scientific #15212-012) was added to the media for 2 additional hours and cells were harvested as follows: Growth medium was stored in a conical tube. Cells were treated twice for 5 min with 1 mL of trypsin. The growth medium and the 2 mL of trypsinization incubations were centrifuged ($250 \times g$ 5 min, 4 °C). Cells were then washed with PBS and resuspended in 75 mM KCl for 15 min at 37 °C. Cells were centrifuged again, the supernatant was removed, and cells were fixed by drop-wise addition of 1 mL fixative (ice-cold methanol: acetic acid, 3:1) while gently vortexing. An additional 9 mL of fixative was then added, and cells were fixed at 4 °C for at least 16 h. Once fixed, metaphases were dropped on glass slides and air-dried overnight. To visualize mitotic cells, slides were mounted in a DAPI-containing ProLong Gold mounting medium (Invitrogen, #P36930). Images were captured on a Zeiss LSM780 laser-scanning confocal.microscope with ZEN 2.3 SP1 software

## Statistical analysis

In vitro studies were performed using at least 3 biological replicates per sample and 3 independent experiments. Two-tailed unpaired $t$ tests were used when comparing two groups. One-way ANOVA followed by Tukey's post hoc comparison was performed for multiple group comparisons. $p < 0.05$ was considered statistically significant. Drug interaction between RP-6306 and RP-3500 was analyzed using the coefficient of drug interaction (CDI)[56]. CDI = AB/(AxB); AB is the ratio of a two-drug combination group to control, and A or B is the ratio of a single drug to control. CDI < 1 indicates synergism, CDI < 0.7 indicates significant synergism, CDI = 1 indicates additivity, and CDI > 1 indicates antagonism. GraphPad Prism (GraphPad Software version 10.0.2, San Diego CA) was used for statistical analyses.

For statistical power for in vivo studies, there were 4-10 mice/arm. Weekly ultrasound measurements, weights, and condition scores were obtained. Longitudinal analysis of tumor growth was carried out by linear mixed-effect modeling with type II ANOVA and pairwise comparisons across groups on log pre-processed tumor sizes using the TumGrowth web tool (https://kroemerlab.shinyapps.io/TumGrowth/)[57] Natural log-transformed tumor volume was used to better satisfy normal distribution. Survival data was analyzed by the Mantel-Cox log-rank test. Survival data was analyzed by the Mantel-Cox log-rank test.

## Reporting summary

Further information on research design is available in the Nature Portfolio Reporting Summary linked to this article.

## Data availability

All the data supporting the findings in this work are included in the main article, supplementary information, or source data file. Source data are provided in this paper.

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

## Acknowledgements

The authors thank the Penn Cytomics and Cell Sorting Shared Resource Laboratory and the Stem Cell & Xenograft Core (SCXC) at the University of Pennsylvania for their support in providing support to flow cytometry and animal studies and M. Hasegan at the Network Biology Collaborative Center for microscopy support. This work was supported by NIH (5R37CA215436-06 (F.S.), 5-P50-CA-228991-04 SPORE and CEP in ovarian cancer (F.S. and H.X.), 1U54CA283759-01 (F.S.), 1 R50CA283807-01A1 (H.X.)); Foundation for Women's Cancer (H.X.), REPARE Therapeutics institutional grant for research.

## Author contributions

Conception and design: F.S., G.M., H.X., E.G., and D.G. Development of methodology: F.S., G.M., H.X., E.G., D.G., S.M., and M.L.H. Acquisition of data (provided animals, acquired and managed patients, provided facilities, etc.): H.X., E.G., D.G., S.M., X.W., R.K., J.F., R.S., E.A., A.P., S.Y., A.S., A.D., H.K., and F.L. Analysis and interpretation of data (e.g., statistical analysis, biostatistics, computational analysis): F.S., G.M., E.G., D.G., S.M., E.A., A.P., and M.L.H. Writing, review, and/or revision of the manuscript: F.S., G.M., H.X., E.G., D.G., S.M., X.W., R.K., J.F., R.S., E.A., A.P., S.Y., A.S., H.K., F.L., M.L.H., A.D., and R.G. Administrative, technical, or material support (i.e., reporting or organizing data, constructing databases): H.X., E.G., D.G., and S.M. Study supervision: F.S. and G.M.

## Competing interests

F.S. serves on scientific advisory boards for AstraZeneca, GSK, and Zentalis Pharmaceuticals. She has received institutional research funding from AstraZeneca, Repare Therapeutics, Instill Bio, and Sierra Oncology. R.K., J.F., R.S., E.A., S.Y.Y., C.G.M. are, and D.G., M.L.H., A.P., A.S. were employees of Repare Therapeutics. R.A.G. serves on the scientific advisory board for Dong-A ST. All other authors declare no competing interests.
