## [Transparent Peer Review file · Nature Communications]

Targeting CCNE1 amplified ovarian and endometrial cancers by combined inhibition of PKMYT1 and ATR

Corresponding Author: Dr Fiona Simpkins

Version 0:

Reviewer comments:

Reviewer #1

(Remarks to the Author)

Xu et al. report their findings on the PKMYT1 inhibitor (PKMYT1i) lunresertib and the ATR inhibitor (ATRi) camonsertib combination as a potential treatment strategy for CCNE1-amplified ovarian and endometrial cancers. The authors demonstrate that PKMYT1i and ATRi synergizes in CCNE1-amplified ovarian and endometrial cancer cell lines and mouse models. Mechanistically, the combination treatment induces CDK1 activation, resulting in premature entry into mitosis, DNA damage and apoptosis in CCNE1-amplified cells.

Of note, the authors previously reported that low doses of WEE1i and ATRi synergize in CCNE1-amplified ovarian and endometrial cancer preclinical models by demonstrating that WEE1i and ATRi combination increases replication fork collapse, RPA exhaustion, and DNA damage in a Cyclin E-CDK2 specific context (Xu, H. et al. Cell Rep Med, 2021). In the present study, the authors show that the mechanisms of action of PKMYT1i and ATRi differ from the mechanisms by WEE1i/ATRi. While PKMYT1i has little effect on CDK2, ATRi further supports PKMYT1i in hyperactivating CDK1 in S phase for CCNE1 amplified cells, thus forcing S phase cells to enter into mitosis, resulting in catastrophic DNA damage.

Overall, the manuscript is well written and provides comprehensive preclinical data of supporting the clinical development of this new drug combination for CCNE1 amplified-ovarian and endometrial cancers. Some clarification and updated data would benefit the current manuscript.

Specific comments are listed below:

1. Experimental design and drug concentrations

1) PKMYT1i has shown synthetic lethality in the CCNE1 overexpressing cells (Gallo, D. et al. 2022). But emergence of resistance to monotherapy is inevitable and toxicity is a main issue when combined with the other DNA repair inhibitors. Thus, combination therapy by using low but active drug concentrations is an advantage for the clinical development. As such, the authors claim that "Our current studies identified a strong synergistic interaction between low doses of PKMYT1i-ATRi in CCNE1-overexpressing/CCNE1-amplified preclinical models" (page 9).

In the current manuscript, the low doses were used for initial cell growth assays (PKMYT1i: 31.3 nM and ATRi:6.25 nM) then doses were escalated for mechanistic studies, by 8-16-fold with maintained 5:1 ratio (PKMYT1i: 250 or 500 nM, ATRi: 50 or 100 nM), which is understandable for 24-72 hours experiments. However, to support the authors' claim, at least DNA damage and apoptosis endpoints using consistent, low doses (PKMYT1i: 31.3nM, ATRi: 6.25nM) should be tested to support the proposed mechanisms.

2) Please define clinically relevant concentrations of lunresertib and camonsertib.

2. The current Methods section lacks details.

1) What was the copy number (CN) cutoff used to define CCNE 1 amplification? Also, clarify the methods used to measure the CCNE1 CN or add references of CCNE1 CN in the cell lines and PDX models used in this study.

2) For cell viability assay used to test drug activity in organoids, was growth inhibition assessed based on the number or size of organoids with drug treatments?

3) The information on the dilution of antibodies used for immunoblotting, flow cytometry, and apoptosis assays is missing.

- 4) Was the coefficient of drug interaction (CDI) based on MTT or colony forming assay results?
- 5) A statistical test must be provided for the correlation data between CCNE1 CN and CDI.

3. The current Results section needs clarification and supporting data.

1) Figure 1:

a. Fig 1D: Not all cell lines tested in Fig 1 A/B were included in this correlation analysis. Please include KLE, KURAMOCHI, MFE280, SKOV3, and SNU685 for the completeness of data. Also, clarify the reason why the particular red dot points were selected for CDI in the correlation analysis.

b. Figure 1E & S1B, 2H-I & S2B:

Please repeat the experiments since colony assays show detached cells in the center (e.g., OVCAR3, KLE, COV318) and periphery (e.g., OVCAR8, OVSAHO) of the wells, likely due to overconfluence and cell detachment prior to fixation and staining. Moreover, uneven cell distribution and aggregation at the edges of the wells indicate insufficient culture medium during cell seeding, which could significantly affect the accuracy of plate quantification.

Additionally, please provide quantification of colony-forming assays. The effects of the combination of PKMYT1i and ATRi appear to be additive rather than synergistic in colony-forming assays (e.g., FUOV1, KLE, OVSAHO).

c. Figure 1F:

It will be important to show the association between drug sensitivity (both PKMYT1i and PKMYT1i/ATRi combination) and CCNE1 CN, mRNA expression, or cyclin E protein expression in other patient-derived organoids in addition to WO-58 to support the claim of CCNE1 amplification as a predictive biomarker.

2) Figure 2:

It is possible that the observed premature mitotic entry induced by the combination may be due to increased low molecular weight cyclin E (LMW-E) rather than the full-length cyclin E, since LMW-E is associated with faster mitotic exit and increased mitotic slippage (Bagheri-Yarmand et al., *Cancer Res*, 2010). Providing the baseline status of both full-length cyclin E and LMW-E alongside CCNE1 amplification would help address this question.

3) Figure 3:

a. The cell line findings suggest the potential of cyclin E1 protein overexpression as a predictive biomarker. To better understand whether the activity of the combination depends on CCNE1 CN or protein overexpression in vivo, PDX models with low CCNE1 CN, but high protein expression should be tested along with CCNE1 amplified PDX models.

b. Fig 3 D Survival curve is missing for OVCAR3.

4) Figure 4:

a. The current analysis covers 2-48 hours for pan-nuclear gamma-H2AX (Figure 4A-E) and 24-72 hours for apoptosis (Figure 4G-J). It would be important to include the same early time points for both assays and apoptosis inhibitors as controls because 1) pan-nuclear gamma-H2AX staining indicates DNA damage as well as increased apoptosis (Rogakou et al., *JBC*, 2000) and 2) both PKMYT1 and ATR inhibit apoptosis through activation of the MAPK pathway (Zhang et al., *Onco Targets Ther*, 2020; Im et al., *JBC*, 2008) or inhibition of apoptotic proteins (Zhang et al., *Cancer Manag Res*, 2019; Hilton et al., *Mol Cell*, 2015).

b. Figure 4F & 6E: Please provide the IHC staining of gamma-H2AX and pCDK1 in other CCNE1-amplified PDX models used in this study (specifically, other PDX models shown in Figure 3) besides OVCAR3.

5) Figure 5:

a. The authors should perform DNA fiber assays to investigate the changes of replication fork dynamics induced by PKMYT1i and ATRi. This information will help better understand whether replication fork breakage or replication catastrophe is or is not a primary mechanism of the action for this combination in CCNE1-amplified cells.

b. To better characterize the effect of the combination on unscheduled mitotic entry, time-lapse microscopy experiment of cells expressing a PCNA chromobody should be performed.

c. Please provide the data on chromosome pulverization and micronuclei in CCNE1-amplified cells induced by combination treatment, as premature mitosis can lead to increased chromosome pulverization and micronuclei formation due to segregation errors during mitosis (Di Bona and Bakhoun, *Cancer Discov*, 2024).

6) Figure 6:

a. To further support that CDK1 activity is essential for combination-induced DNA damage and premature mitosis entry, please include the CDK1 inhibitor RO-3306 as a control.

b. The authors have reported several genes related to CCNE1, such as MYBL2, LIN54, FOXM1, and FBXW7, that are associated with sensitivity to PKMYT1i (Gallo et al., *Nature*, 2022). Do the models used in this study have alterations in these genes?

7) Discussion:

a. Please revise the discussion as it largely repeats the results, especially the 2nd and 3rd paragraphs.

b. Please include the limitations of this study.

c. The authors should also discuss other possible biomarkers for predicting the response of the combination. For instance, LMW-E positive status (Lulla et al., 2023 AACR meeting; Chen et al., *Cancers*, 2021; Li et al., *Oncogene*, 2022) or the CCNE1-related genes reported by the authors (Gallo et al., *Nature*, 2022; Hu et al., *Cell Rep Med*, 2021).

Minor comments:

1. Figure 1A-B: The summary table on the right displays the IC50 values for each drug. For better readability, please provide the exact copy number of each cell line and explain the meaning of the fold shift listed below in the figure legend.

2. Statistical tests should be included in all legends.
3. Please define the red dot in Figure 1D. Also, please include the names of the cell lines in the correlation graph for clarity.
4. Figure 1C, 2E, 3A-D, 5B-C, 6F, 9B: Please provide the significance between RP-3500 (or RP-6306) and combination.
5. Figure 1F: Please provide the scale bar for the images.
6. All quantification results of western blot analysis should be provided in the Supplementary Figures.

Reviewer #2

(Remarks to the Author)

In their manuscript entitled: 'Targeting CCNE1 amplified ovarian and endometrial cancers by combined inhibition of PKMYT1 and ATR', Xu and co-workers describe the combined inhibition of two cell cycle checkpoint inhibitors. Cyclin E1 amplification describes a class of hard-to-treat cancers, and efforts to improve treatment for these patients is applauded. Previous efforts, including studies by these authors have already investigated cell cycle checkpoint inhibitors. The current combination of inhibitors of PKMYT1 and ATR that are already clinically evaluated could guide future clinical trials.

This is very straightforward study, and its value is that the tested treatment could easily be tested in the clinic. The synergistic effects in vitro are most pronounced in the isogenic models in contrast to the panels of patient-derived models. The in vivo effects, particularly in WU-155, are impressive. Mechanistically, all the effects that were previously seen upon PKMYT1 inhibition as single agent, are more pronounced upon combined PKMYT1/ATR inhibition. The only mechanistic new finding is that PKMYT1 inhibition actually triggers ATR activity (as judged by Chk1 phosphorylation), but this finding is not followed up in depth.

Comments

1. I find the synergistic effects not always very clear, especially in the patient-derived models. The isogenic models (FT282 and SNU685) show clearer results. Can the authors speculate on these differences (perhaps due to adaptation during tumorigenesis?).
2. CCNE1 amplified cancers were previously shown to be sensitive to ATR, WEE1 and PKMYT1 inhibitors, as well as combined ATR-WEE1 inhibition. Especially considering the combined ATR-WEE1i effects, you could argue that the effects of the PKMYT1-ATR combination is not very surprising. Can the authors speculate on the various combinations, and why ATR-PKMYT1 is different/superior?
3. Figure 1A: The PKMYT1 is reported to be effective in CCNE1 amp cells, with limited activity in CCNE1 gain cell lines, and no activity in CCNE1 neutral cells. It seems that 2 of the CCNE1 amp models (FUOV1 and COV318 do now show activity and cluster with the gain/neutral models. This should be more accurately described.
4. Similarly, it is mentioned that ATRi does not have clear single agent activity, but in Fig. 1B, all CCNE1 amp cell lines cluster together with higher sensitivity. The clustering is actually more impressive when compared to PKMYT1 single agent. Again here, the results section should better reflect the data. (also in Fig 1C, the single agent activity for ATRi seems more impressive when compared to PKMYT1i).
5. The clonogenic assays (Fig 1E) do not show impressive synergistic effects, and should be quantified. Same for the WO-38 organoid lines: are these effects synergistic or additive?
6. The WO-58 is described to have a BRCA1 mutation and CCNE1 amp. these has previously been described as mutually exclusive genetic events. Is this model HRD?
7. Figure 2H-I: clonogenic assays require quantifications over biological replicates.
8. Figure 6C: PKMYT1 treatment leads to ATR/CHK1 activation, this should be part of the model in Figure 7. Also, the degree to which ATR/CHK1 is activated upon PKMYT1 inhibition may be a good indication of the efficacy of the combination treatment. Would be good to see if Chk1-p phosphorylation is consistently seen in the cell line panel after PKMYT1 inhibition.

Reviewer #3

(Remarks to the Author)

REVIEWER COMMENTS

In this work Xu and collaborators investigate the therapeutic role of PKMYT1 inhibitor RP-6306 with the ATR inhibitor RP-3500 combination on endometrial and ovarian carcinomas. Although the work is well conducted and potentially interesting as a novel therapeutic approach for highly aggressive endometrial and ovarian malignancies, there are some issues that I think that should be addressed.

General point:

One my main and general concerns is related to the clinical relevance of author's findings. Although CCNE1 overexpressing EMCA account for 5-6% of all endometrial neoplasia, they are high grade cancer, with aggressive behavior and metastatic potential. From this point of view, the proposed combinatorial treatment, can be a new therapeutic approach for these patients. However, metastatic outgrowth is the main responsible for poor prognosis of these type of tumors. For this reason, It would be very interesting to analyze the effects of RP-6306 and RP-3500 combination in metastases of EMCA and OVCA.

Specific points:

1. In figure 1, the authors use OVCA (OVCAR3, FUOV1, COV318, OVCAR8, OVSAHO, Kuramochi, WO-20, SKOV3) and

EMCA (KLE, MFE280, SNU685) cell lines that are classified in three groups on basis of CCNE1 copy numbers as CCNE1 amplified (>5), CCNE1 gain (2-5) or CCNE1 low (neutral, I understand 2 copies). However, they do not show whether such copy number variation correlate with expression of CCNE1 at mRNA and protein level. This correlation must be provided.

2. In figure 1, the authors correlate the effects of RP-6306 and RP-3500 on cell viability of EMCA and OVCA cell lines with the copy number variations of CCNE1 and they clearly show synergistic effect of the tested drugs. However, most of cancer cell lines display a high burden of molecular alterations. Therefore, CCNE1 amplification is one molecular alteration among hundreds of them and CCNE1-amplified cell lines may share other molecular alterations that may be implied RP-6306 and RP-3500 sensitivity. Although in Figure 2, the authors show that CCNE1 overexpression increases synergistic effects of RP-6306 and RP-3500 there is no demonstration that reduction of CCNE1 in cells harboring high copy numbers of CCNE1 reduces sensitivity to the drugs. To sort this out, I would recommend reducing expression of CCNE1 in amplified cell lines (using shRNAs or CRISPR editing) and compare the effects of the drugs with parental cell lines.

3. It is not clear how cell viability is assessed in Figure 1F. Could it be clarified? Is the reduction of cell viability caused by apoptotic cell death in this case? Or other type of cell death? This point should be addressed.

4. In Figure 3, the authors clearly show a synergistic effect ofc in PDX models of OVCA and EMCA. In all PDX, combination therapy significantly delays progression of disease. However, after several weeks of treatment (depending on PDX), tumors start to grow again reaching volumes above 1000 mm. This is probably caused by acquired resistance to treatment. Theoretically, clones lacking CCNE1 amplification or clone, but other molecular alterations would be responsible for such resistance. Alternatively, clones of cells containing CCNE1 AMP, acquire other mutations conferring resistance to treatment. Therefore, from the clinical point of view, it would be very interesting to see whether resistant tumors that progress after treatment display CCNE1 amplification. At the molecular level, it would be interesting to see whether these tumors activate apoptosis and H2AX during regression period, but no when they acquire resistance to treatment.

5. Mechanistically, the authors show marked increase of H2AX phosphorylation when combining ATR and PKMTY1 inhibitors in all tested cell lines, but this increase is not observed when cells are treated with RP-3500 alone in any of the cells analyzed. Being RP-3500 an inhibitor of ATR, I would expect an increase of DNA damage when used as single treatment. How do the authors explain this observation? Moreover, in Figure 4A it seems that RP-6306 FT282 WT cells display a significant increase of H2AX phosphorylation that is not observed CCNE1/OE FT282 cells. Does it mean that CCNE1 OE reduces DNA damage after inhibition of PKMTY1?. Is this observed in other cell types?

6. In figure 6I-F, a cleaved caspase-3 demonstrates that combination of the two drugs enhances apoptotic cells death. However, in KLE cells without any treatment there a marked increase of caspase-3 after 48 hours is observed. Although it might be due to unequal protein loading, I think that these western should be improved and additional markers of apoptosis should be shown (annexin V, caspase-9, PARP). I would also recommend showing a immunohistochemistry of cleaved caspase-3 in OVAR3 xenotransplants (as shown for H2AX).

7. Finally, it is well known that pharmacological inhibitors are not completely specific, and they target other molecules besides the ones that have been designed for. For this reason, it would be interesting to demonstrate that genetic inhibition of ATR and PKMTY1 inhibition has similar effects to those observed with the inhibitors. These experiments wouldn't rule out the possible effects on other targets, but at least they would demonstrate that the anti-cancer effects are caused by inhibition of the expected targets.

Version 1:

Reviewer comments:

Reviewer #1

(Remarks to the Author)

The authors have substantially improved the manuscript with new experiments though in-depth mechanistic insights are still lacking for this combination.

1) comment on Figure 4G: The gamma-H2AX intensity in the RP-6306 representative image appears higher than in the combination treatment image, which seems inconsistent with the quantification data presented in the bar graph. Please carefully review the representative images and verify the accuracy of the quantification process.

Reviewer #2

(Remarks to the Author)

the authors have adequately commented on the issues that were raised. overall the manuscript has improved, and I support publication.

Reviewer #3

(Remarks to the Author)

Although the authors have addressed and solved most of my concerns, there still are two important points that need to be

further clarified.

1. In my first review I expressed my main concern about the significance of the results:

“My main and general concern is related to the clinical relevance of author’s findings. Although CCNE1 overexpressing EMCA account for 5-6% of all endometrial neoplasia, they are high grade cancer, with aggressive behavior and metastatic potential. From this point of view, the proposed combinatorial treatment can be a new therapeutic approach for these patients. However, metastatic outgrowth is the main responsible for poor prognosis of these type of tumors. For this reason, It would be very interesting to analyze the effects of RP- 6306 and RP-3500 combination in metastases of EMCA and OVCA.”

The authors answered my question as follows.

- Metastatic ovarian and endometrial cancers are indeed a major clinical challenge us physicians face when treating these patients. Two of 3 PDX models tested (WO-77, WO-19) developed tumor metastasis and all develop ascites which were significantly reduced with the combination making this combination clinically relevant.

However, I am unable to find the results regarding the effects of drug combination on metastatic outgrowth of the above-mentioned PDX. Could the authors clarify in which figures are shown the effects on metastatic spreads?

Moreover, I am unable to find the word metastasis in any section of the results nor in the figure legends, where are these results explained?

2. To my previous question:

“In Figure 3, the authors clearly show a synergistic effect ofc in PDX models of OVCA and EMCA. In all PDX, combination therapy significantly delays progression of disease. However, after several weeks of treatment (depending on PDX), tumors start to grow again reaching volumes above 1000 mm. This

It is probably caused by acquired resistance to treatment. Theoretically, clones lacking CCNE1 amplification or clone, but other molecular alterations would be responsible for such resistance. Alternatively, clones of cells containing CCNE1 AMP , acquire other mutations conferring resistance to treatment. Therefore, from the clinical point of view, It would be very interesting to see whether resistant tumors that progress after treatment display CCNE1 amplification. At the molecular level, it would be interesting to see whether these tumors activate apoptosis and H2AX during regression period, but no when they acquire resistance to treatment.

The authors answered my question as follows.

- We thank the reviewer for this observation and agree that this would be an interesting hypothesis to test. We plan to look into mechanisms of resistance in future studies.

I understand the authors will study in depth the molecular mechanisms of resistance and these results are beyond the scope of the present work. However, I think that these observations about resistance should be properly stated and discussed.

Version 2:

Reviewer comments:

Reviewer #3

(Remarks to the Author)

The authors have addressed all my main concerns an questions.

Point-by-point response

REVIEWER #1:

Xu et al. report their findings on the PKMYT1 inhibitor (PKMYT1i) lunresertib and the ATR inhibitor (ATRi) camonsertib combination as a potential treatment strategy for CCNE1-amplified ovarian and endometrial cancers. The authors demonstrate that PKMYT1i and ATRi synergizes in CCNE1-amplified ovarian and endometrial cancer cell lines and mouse models. Mechanistically, the combination treatment induces CDK1 activation, resulting in premature entry into mitosis, DNA damage and apoptosis in CCNE1-amplified cells. Of note, the authors previously reported that low doses of WEE1i and ATRi synergize in CCNE1-amplified ovarian and endometrial cancer preclinical models

by demonstrating that WEE1i and ATRi combination increases replication fork collapse, RPA exhaustion, and DNA damage in a Cyclin E-CDK2 specific context (Xu, H. et al. Cell Rep Med, 2021). In the present study, the authors show that the mechanisms of action of PKMYT1i and ATRi differ from the mechanisms by WEE1i/ATRi. While PKMYT1i has little effect on CDK2, ATRi further supports PKMYT1i in hyperactivating CDK1 in S phase for CCNE1 amplified cells, thus forcing S phase cells to enter into mitosis, resulting in catastrophic DNA damage.

Overall, the manuscript is well written and provides comprehensive preclinical data of supporting the clinical development of this new drug combination for CCNE1 amplified-ovarian and endometrial cancers. Some clarification and updated data would benefit the current manuscript.

Specific comments are listed below:

1. Experimental design and drug concentrations

1) PKMYT1i has shown synthetic lethality in the CCNE1 overexpressing cells (Gallo, D. et al. 2022). But emergence of resistance to monotherapy is inevitable and toxicity is a main issue when combined with the other DNA repair inhibitors. Thus, combination therapy by using low but active drug concentrations is an advantage for the clinical development. As such, the authors claim that "Our current studies identified a strong synergistic interaction between low doses of PKMYT1i-ATRi in CCNE1-overexpressing/CCNE1-amplified preclinical models" (page 9).

In the current manuscript, the low doses were used for initial cell growth assays (PKMYT1i: 31.3 nM and ATRi:6.25 nM) then doses were escalated for mechanistic studies, by 8-16-fold with maintained 5:1 ratio (PKMYT1i: 250 or 500 nM, ATRi: 50 or 100 nM), which is understandable for 24-72 hours experiments. However, to support the authors' claim, at least DNA damage and apoptosis endpoints using consistent, low doses (PKMYT1i: 31.3nM, ATRi: 6.25nM) should be tested to support the proposed mechanisms.

- We now tested the low dosage combination (PKMYT1i: 31.3nM, ATRi:6.25nM) in OVCAR3 and FT282 cells. The low dosage combination increased DNA damage and induced cell apoptosis in OVCAR3 cells and FT282 cells with CCNE1 overexpression. Similar effects were seen at lower dose to higher doses, but at later time points. (Revised Fig. S6E, S7).

2) Please define clinically relevant concentrations of lunresertib and camonsertib.

- Pharmacokinetic (PK) analysis of combined lunresertib/camonsertib treatment in humans at the recommended phase 2 dose (RP2D) is currently being explored as part of the MYTHIC clinical

trial. Data analysis is still ongoing and thus, it is premature to introduce the clinically relevant dose in the manuscript. We have removed these statements from the manuscript. The blood concentrations of lunresertib and camonsertib at RP2D in humans will be included in a future MYTHIC study publication.

2. The current Methods section lacks details.

1) *What was the copy number (CN) cutoff used to define CCNE 1 amplification? Also, clarify the methods used to measure the CCNE1 CN or add references of CCNE1 CN in the cell lines and PDX models used in this study.*

- We thank the reviewer for this comment and agree that more information about CCNE1 CN cutoff is required. In the revised manuscript methods section we defined the cutoff for amp vs. gain vs. neutral/low and added references for genomic characterization of established cell lines (Cancer Cell Line Encyclopedia) and PDX models¹.

2) *For cell viability assay used to test drug activity in organoids, was growth inhibition assessed based on the number or size of organoids with drug treatments?*

- Organoid viability was tested after drug treatment for indicated times using the Cell Counting Kit-8 (CCK-8) assay. We didn't measure Numbers of organoids or size of organoids in the cell viability assay.

3) *The information on the dilution of antibodies used for immunoblotting, flow cytometry, and apoptosis assays is missing.*

- We thank the reviewer for pointing out this omission, all antibody dilutions are included in the methods section of the revised manuscript.

4) *Was the coefficient of drug interaction (CDI) based on MTT or colony forming assay results?*

- CDI was calculated in Figure 1C using the MTT assay, which is now indicated in figure legend 1 of the revised manuscript.

5) *A statistical test must be provided for the correlation data between CCNE1 CN and CDI.*

- We have conducted a simple linear regression for the correlation between CCNE1 CN and CDI which is indicated in figure legend 1 of the revised manuscript.

3. The current Results section needs clarification and supporting data.

1) **Figure 1:**

a. **Fig 1D:** *Not all cell lines tested in Fig 1 A/B were included in this correlation analysis. Please include KLE, KURAMOCHI, MFE280, SKOV3, and SNU685 for the completeness of data. Also, clarify the reason why the particular red dot points were selected for CDI in the correlation analysis.*

- Combination studies were only done in the indicated cell lines, which is why KLE, KURAMOCHI, MFE280, SKOV3, and SNU685 were not included in the correlation analysis. We included the SNU685 CCNE1 inducible line in Fig. 2D to demonstrate the dependence of this drug combination on CCNE1 levels. We clarified why the red dot points were selected in the figure legends.

b. **Figure 1E & S1B, 2H-I & S2B:**

Please repeat the experiments since colony assays show detached cells in the center (e.g., OVCAR3, KLE, COV318) and periphery (e.g., OVCAR8, OVSAHO) of the wells, likely due to over confluence and cell detachment prior to fixation and staining. Moreover, uneven cell distribution and aggregation at the edges of the wells indicate insufficient culture medium during cell seeding, which could significantly affect the accuracy of plate quantification. Additionally, please provide quantification of colony-forming assays. The effects of the combination of PKMYT1i and ATRi appear to be additive rather than synergistic in colony-forming assays (e.g., FUOV1, KLE, OVSAHO).

- We did repeat all colony formation experiments and they show similar results. Cell number plated is normalized by doubling time so results are comparable between cell lines. Cells are treated at low confluence and media is sufficient as it is refreshed every 3 days. OVCAR3, KLE and OVSAHO images are updated in the current figure (Revised Fig. 1E). As requested by the reviewer, we have now added quantification of all colony formation assays using Image J in the revised manuscript.. The Coefficient of drug interaction (CDI) was also calculated and added to show the synergy of the combination in these lines. (Revised Fig. 1F, Fig. S1C).

c. **Figure 1F:**

It will be important to show the association between drug sensitivity (both PKMYT1i and PKMYT1i/ATRi combination) and CCNE1 CN, mRNA expression, or cyclin E protein expression in other patient-derived organoids in addition to WO-58 to support the claim of CCNE1 amplification as a predictive biomarker.

- In addition to WO-58 organoids, we now have tested the PKMYT1i/ATRi combination in WO-19 and WO-77 organoids, two additional CCNE1 amplified organoids in revised manuscript. The PKMYT1i-ATRi combination showed better growth inhibition in these two CCNE1 amplified organoid models. (Revised Fig. 1H).

2) Figure 2:

It is possible that the observed premature mitotic entry induced by the combination may be due to increased low molecular weight cyclin E (LMW-E) rather than the full-length cyclin E, since LMW-E is associated with faster mitotic exit and increased mitotic slippage (Bagheri-Yarmand et al., Cancer Res, 2010). Providing the baseline status of both full-length cyclin E and LMW-E alongside CCNE1 amplification would help address this question.

- We appreciate the reviewer's comments regarding differences in LMW-E versus full-length cyclin E and agree that the baseline status of both full-length cyclin E and LMW-E may be important in the observed premature mitotic entry induced by the combination. We have previously identified by full length and low molecular weight isoforms in the Cyclin E1 overexpression cell lines used in this study¹, which may make it difficult to identify the contribution of one isoform over the other. This was also not a focus of this study. We were interested in identifying a biomarker that would be readily accessible in the clinics for rapid clinical translation of this combination.

3) Figure 3:

a. The cell line findings suggest the potential of cyclin E1 protein overexpression as a predictive biomarker. To better understand whether the activity of the combination depends on CCNE1 CN or protein overexpression in vivo, PDX models with low CCNE1 CN, but high protein expression should be tested along with CCNE1 amplified PDX models.

[Redacted]

[Redacted]

b. Fig 3 D Survival curve is missing for OVCAR3.

- Unfortunately, the OVCAR3 efficacy study was not designed to measure survival since treatments were stopped after day 25. For this reason, we have not included the OVCAR3 survival data in the manuscript.

4) Figure 4:

a. The current analysis covers 2-48 hours for pan-nuclear gamma-H2AX (Figure 4A-E) and 24-72 hours for apoptosis (Figure 4G-J). It would be important to include the same early time points for both assays and apoptosis inhibitors as controls because 1) pan-nuclear gamma-H2AX staining indicates DNA damage as well as increased apoptosis (Rogakou et al., JBC, 2000) and 2) both PKMYT1 and ATR inhibit apoptosis through activation of the MAPK pathway (Zhang et al., Onco Targets Ther, 2020; Im et al., JBC, 2008) or inhibition of apoptotic proteins (Zhang et al., Cancer Manag Res, 2019; Hilton et al., Mol Cell, 2015).

[Redacted]

b. Figure 4F & 6E:

Please provide the IHC staining of gamma-H2AX and pCDK1 in other CCNE1-amplified PDX models used in this study (specifically, other PDX models shown in Figure 3) besides OVCAR3.

Penn Medicine

University of Pennsylvania School of Medicine

[Redacted]

5) *Figure 5:*

a. *The authors should perform DNA fiber assays to investigate the changes of replication fork dynamics induced by PKMYT1i and ATRi. This information will help better understand whether replication fork breakage or replication*

catastrophe is or is not a primary mechanism of the action for this combination in CCNE1-amplified cells.

- We performed the DNA fiber assay in OVCAR3 with drugs treatment. PKMYT1i and ATRi monotherapy slow down the DNA fiber speed, preventing DNA replications. The combination showed even stronger effect in decreasing DNA fiber progression (Revised Fig. 5D).

b. *To better characterize the effect of the combination on unscheduled mitotic entry, time-lapse microscopy experiment of cells expressing a PCNA chromobody should be performed.*

- We agree this is a perfect experiment to solidify that the combo induces more unscheduled mitosis. We treated PCNA-cb-TagRFP expressing FT282 WT and CCNE1-OE cells⁵ with RP-6306, RP-3500 or the combination and imaged over 48 h (Revised Fig. 5C). We observe that combined RP-6306/RP-3500 treatment increased the number of S-phase nuclear envelope breakdown events in CCNE1-OE cells above that of either single-agent alone. Importantly, the combination had only mild effects in the WT FT282 cells indicating the synthetic lethal window is maintained. This data supports the claim that combined PKMYT1/ATR inhibition increases unscheduled mitotic entry in CCNE1-OE cells.

c. *Please provide the data on chromosome pulverization and micronuclei in CCNE1-amplified cells induced by combination treatment, as premature mitosis can lead to increased chromosome pulverization and micronuclei formation due to segregation errors during mitosis (Di Bona and Bakhoun, Cancer Discov, 2024).*

- This is another great suggestion to bolster the mechanism-of-action. We treated FT282 WT and CCNE1-OE cells with RP-6306, RP-3500 or the combination and performed metaphase spreads to measure chromosome pulverization or performed QIBC to measure micronucleation. In accordance with the unscheduled mitosis and γ H2AX results, we see that the RP-6306/RP-3500 combination increased the percentage of CCNE1-OE cells with chromosome pulverization above that of either single agent or the combination in WT cells (Revised Fig. 4E). This reinforces the claim that unscheduled mitotic entry leads to chromosome pulverization and is the major mechanism of cell death after combined PKMYT1/ATR inhibition in CCNE1-OE cells.
- Combined RP-6306/RP-3500 treatment also increased percent of CCNE1-OE cells with micronucleation above either single-agent alone (Revised Fig. S6F). However, there is a substantial effect of RP-3500 alone suggesting additivity rather than synergy. While the RP-3500

result is not surprising given the roles of ATR in the replication stress response⁶, we believe that chromosome pulverization is major source of genotoxic stress elicited by combined PKMYT1/ATR inhibition and micronucleation is only an intermediate phenotype. Regardless, the increased micronucleation does raise interesting questions about activation of cGAS-STING and the potential for combination IO therapies. We are actively investing this avenue from both a mechanistic and therapeutic perspective.

6) *Figure 6:*

a. *To further support that CDK1 activity is essential for combination-induced DNA damage and premature mitosis entry, please include the CDK1 inhibitor RO-3306 as a control.*

- Thank you for this suggestion. To address this, we treated FT282 CCNE1-OE cells with RP-6306/RP-3500 plus increasing concentrations of RO-3306 and measured the percentage of cells with pan-gH2AX (Revised Fig. S9F). RO-3306 significantly reduced the amount pan-gH2AX indicating the majority of DNA damage induction is CDK1-dependent. The residual DNA damage that is CDK1-independent likely arises from small amount of replication catastrophe we observe in RP-6306/RP-3500 treated CCNE1-OE cells (Revised Fig. S9F)

b. *The authors have reported several genes related to CCNE1, such as MYBL2, LIN54, FOXM1, and FBXW7, that are associated with sensitivity to PKMYT1i (Gallo et al., Nature, 2022). Do the models used in this study have alterations in these genes?*

- The mutations for the PDXs have been summarized in the supplemental figures. They do not have any mutations associated with sensitivity to PKMYT1i. We reviewed mutations in DepMap and checked ClinVar for pathogenicity for the cell lines used in our paper. We found that KLE and SKOV3 have likely pathogenic FBXW7 mutations and OVCAR3 has a PPP2R1A mutation that is not characterized in ClinVar yet (p.E100K).

7) *Discussion:*

a. *Please revise the discussion as it largely repeats the results, especially the 2nd and 3rd paragraphs.*

- We readdressed the discussion section to decrease repetitiveness of the results

b. *Please include the limitations of this study.*

- We added a paragraph to the discussion regarding study limitations.

c. The authors should also discuss other possible biomarkers for predicting the response of the combination. For instance, LMW-E positive status (Lulla et al. 2023 AACR meeting; Chen et al., *Cancers*, 2021; Li et al., *Oncogene*, 2022) or the CCNE1-related genes reported by the authors (Gallo et al., *Nature*, 2022; Hu et al., *Cell Rep Med*, 2021).

- We appreciate the reviewer's comments and inclusion of references. We agree that there may be other biomarkers predictive of response to the combination and have added some comments and references about low molecular weight Cyclin E1 and other CCNE1-related genes to the discussion section.

Minor comments:

1. Figure 1A-B: The summary table on the right displays the IC50 values for each drug. For better readability, please provide the exact copy number of each cell line and explain the meaning of the fold shift listed below in the figure legend.

- We added the copy number of each cell lines as defined in DepMap and added a description of fold shift to the figure legend for Figure 1A-B.

2. Statistical tests should be included in all legends.

- Statistical tests have been added to all legends.

3. Please define the red dot in Figure 1D. Also, please include the names of the cell lines in the correlation graph for clarity.

- A description of the red dot has been added to Figure 1D. The names of the cell lines were added to the correlation graph.

4. Figure 1C, 2E, 3A-D, 5B-C, 6F, 9B: Please provide the significance between RP-3500 (or RP-6306) and combination.

- The significance between monotherapy and combination has been added to the indicated figures.

5. Figure 1F: Please provide the scale bar for the images.

- The scale bar is included.

6. All quantification results of western blot analysis should be provided in the Supplementary Figures.

- The quantification of western blot assay is included.

REVIEWER #2

In their manuscript entitled: 'Targeting CCNE1 amplified ovarian and endometrial cancers by combined inhibition of PKMYT1 and ATR', Xu and coworkers describe the combined inhibition of two cell cycle checkpoint inhibitors. Cyclin E1 amplification describes a class of hard-to-treat cancers, and efforts to improve treatment for these patients is applauded. Previous efforts, including studies by these authors have already investigated cell cycle checkpoint inhibitors. The current combination of inhibitors of PKMYT1 and ATR that are already clinically evaluated could guide future clinical trials.

This is very straightforward study, and its value is that the tested treatment could easily be tested in the clinic. The synergistic effects in vitro are most pronounced in the isogenic models in contrast to the panels of patient-derived models. The in vivo effects, particularly in WU-155, are impressive. Mechanistically, all the effects that were previously seen upon PKMYT1 inhibition as single agent, are more pronounced upon combined PKMYT1/ATR inhibition. the only mechanistic new finding is that PKMYT1 inhibition actually triggers ATR activity (as judged by Chk1 phosphorylation), but this finding is not followed up in depth.

Comments

1. *I find the synergistic effects not always very clear, especially in the patient derived models. The isogenic models (FT282 and SNU685) show clearer results. can the authors speculate on these differences (perhaps due to adaptation during tumorigenesis?).*

- We added comments in the discussion section to speculate on the difference seen between 2D cell models and in vivo models. There are limitation to 2D culture, which does not fully recapitulate the complex cell-cell and cell-environment interactions of a 3D or in vivo system. The cancer's molecular landscape, heterogeneity, and tumor microenvironment all influence tumorigenesis, metastasis, response to treatment, and emergence of drug resistance.

2. *CCNE1 amplified cancers were previously shown to be sensitive to ATR, WEE1 and PKMYT1 inhibitors, as well as combined ATR-WEE1 inhibition. Especially considering the combined ATR-WEE1i effects, you could argue that the effects of the PKMYT1-ATR combination is not very surprising. Can the authors speculate on the various combinations, and why ATR-PKMYT1 is different/superior?*

- This is an important point and we thank the reviewer for raising it. We agree that while the effect of combined PKMYT1-ATRi results is not surprising based on the literature, it does prove this is a suitable combination for clinical development. The success of developing DDR inhibitor combinations in the clinic will hinge on effectively managing toxicity by careful selection of dose/schedule and combination partners. Monotherapy of both WEE1 and ATR inhibitors are

both associated with myelosuppression suggesting an overlapping toxicity profile^{7,8}. Conversely, RP-6306 monotherapy does not lead to substantial myelosuppression (https://aacrjournals.org/mct/article/22/12_Supplement/PR008/730843/Abstract-PR008-MYTHIC-First-in-human-FIH-biomarker) suggesting a more favorable toxicity profile for combination with ATR inhibitors (or WEE1). We added a statement to the discussion about this point.

3. *Figure 1A: The PKMYT1 is reported to be effective in CCNE1 amp cells, with limited activity in CCNE1 gain cell lines, and no activity in CCNE1 neutral cells. It seems that 2 of the CCNE1 amp models (FUOV1 and COV318 do now show activity and cluster with the gain/neutral models. This should be more accurately described.*

- We thank the reviewers for pointing this out and have made adjustments with the wording here as recommended.

4. *Similarly, it is mentioned that ATRi does not have clear single agent activity, but in Fig. 1B, all CCNE1 amp cell lines cluster together with higher sensitivity. The clustering is actually more impressive when compared to PKMYT1 single agent. Again here, the results section should better reflect the data. (also in Fig 1C, the single agent activity for ATRi seems more impressive when compared to PKMYT1i).*

- We agree with the reviewers on this point and have made adjustments with the wording here as recommended.

5. *The clonogenic assays (Fig 1E) do not show impressive synergistic effects, and should be quantified. Same for the WO-38 organoid lines: are these effects synergistic or additive?*

- Thank you for the suggestion. We quantified the clonogenic assay data and calculated the CDI and it is synergistic (Revised Figure 1F, Figure S1C). We also calculated the CDI for organoid experiments. The drugs show synergy in WO-19 and WO-77 organoids. However, in WO-58, both monotherapies significantly inhibited organoid viability. Combination further decreased viability compared to monotherapies, but was not synergistic at the doses tested (Revised Figure S1D).

6. *The WO-58 is described to have a BRCA1 mutation and CCNE1 amp. These has previously been described as mutually exclusive genetic events. Is this model HRD?*

- WO-58 is a BRCA1 mutant model and acquired CCNE1 amplification upon developing resistance to PARP inhibitor. This has been shown in patient samples as a mechanism of PARPi resistance (Evolve study by Stephanie L'Heureux, Clinical Cancer Research, 2020). It has 39% positive RAD51 foci (foci >5) cells, indicating this is a HR proficient model.

7. *Figure 2H-I: clonogenic assays require quantifications over biological replicates.*

- We quantified the clonogenic assays over biological replicates which is now included in the revised manuscript.

8. Figure 6C: PKMYT1 treatment leads to ATR/CHK1 activation, this should be part of the model in Figure 7. Also, the degree to which ATR/CHK1 is activated upon PKMYT1 inhibition may be a good indication of the efficacy of the combination treatment. Would be good to see if Chk1-p phosphorylation is consistently seen in the cell line panel after PKMYT1 inhibition.

- PKMYT1 treatment does indeed lead to CHK1 activation as shown in Figure 6C. Figure 7 now makes this relationship clearer.

REVIEWER #3

In this work Xu and collaborators investigate the therapeutic role of PKMYT1 inhibitor RP-6306 with the ATR inhibitor RP-3500 combination on endometrial and ovarian carcinomas. Although the work is well conducted and potentially interesting as a novel therapeutic approach for highly aggressive endometrial and ovarian malignancies, there are some issues that I think that should be addressed.

General point:

One my main and general concerns is related to the clinical relevance of author's findings. Although CCNE1 overexpressing EMCA account for 5-6% of all endometrial neoplasia, they are high grade cancer, with aggressive behavior and metastatic potential. From this point of view, the proposed combinatorial treatment, can be a new therapeutic approach for these patients. However, metastatic outgrowth is the main responsible for poor prognosis of these type of tumors. For this reason, It would be very interesting to analyze the effects of RP- 6306 and RP-3500 combination in metastases of EMCA and OVCA.

- Metastatic ovarian and endometrial cancers are indeed a major clinical challenge us physicians face when treating these patients. Two of 3 PDX models tested (WO-77, WO-19) developed tumor metastasis and all develop ascites which were significantly reduced with the combination making this combination clinically relevant.

Specific points:

1. *In figure 1, the authors use OVCA (OVCAR3, FUOV1, COV318, OVCAR8, OVSAHO, Kuramochi, WO-20, SKOV3) and EMCA (KLE, MFE280, SNU685) cel llines that are classified in three groups on basis of CCNE1 copy numbers as CCNE1 amplified (>5), CCNE gain (2-5) or CCNE low (neutral, I understand 2 copies). However, they do not show whether such copy number variation correlate with expression of CCNE1 at mRNA and protein level. This correlation must be provided.*

- We added correlation analysis for CCNE1 copy number and mRNA expression as well as CCNE1 copy number and protein level in Supplemental Figure IC.

2. In figure 1, the authors correlate the effects of RP-6306 and RP-3500 on cell viability of EMCA and OVCA cell lines with the copy number variations of CCNE1 and they clearly show synergistic effect of the tested drugs. However, most of cancer cell lines display a high burden of molecular alterations. Therefore, CCNE1 amplification is one molecular alteration among hundreds of them and CCNE1-amplified cell lines may share other molecular alterations that may be implied RP-6306 and RP-3500 sensitivity. Although in Figure 2, the authors show that CCNE1 overexpression increases synergistic effects of RP-6306 and RP-3500 there is no demonstration that reduction of CCNE1 in cells harboring high copy numbers of CCNE1 reduces sensitivity to the drugs. To sort this out, I would recommend reducing expression of CCNE1 in amplified cell lines (using shRNAs or CRISPR editing) and compare the effects of the drugs with parental cell lines.

- We thank the reviewer for this suggestion. We had considered conducting these experiments; however, previous work showing siRNA knockdown of CCNE1 reduced cell growth 65-90% in OVCA cell lines with elevated CCNE1 expression, including OVCAR3 and OVCAR8⁹, suggesting they are oncogene-addicted and depend on cyclin E for survival. We believe it would be difficult to interpret additional growth defects from RP-6306/RP-3500 under these conditions. As the reviewer noted, we have shown that overexpression of CCNE1 in multiple cell lines engenders them to combined PKMYT1/ATR inhibition.

3. It is not clear how cell viability is assessed in Figure 1F. Could it be clarified? Is the reduction of cell viability caused by apoptotic cell death in this case? Or other type of cell death? This point should be addressed.

- The cell viability of organoids was measured with Cell Counting Kit-8 (CCK8) assay. We didn't tested how it induced cell death. It may be through inducing cell apoptosis as we observed in the 2D experiments.

4. In Figure 3, the authors clearly show a synergistic effect ofc in PDX models of OVCA and EMCA. In all PDX, combination therapy significantly delays progression of disease. However, after several weeks of treatment (depending on PDX), tumors start to grow again reaching volumes above 1000 mm. This isprobably caused by acquired resistance to treatment. Theoretically, clones lacking CCNE1 amplification or clone, but other molecular alterations would be responsible for such resistance. Alternatively, clones of cells containing CCNE1 AMP, acquire other mutations conferring resistance to treatment. Therefore, from the clinical point of view, It would be very interesting to see whether resistant tumors that progress after treatment display CCNE1 amplification. At the molecular level, it would be interesting to see whether these tumors activate apoptosis and H2AX during regression period, but no when they acquire resistance to treatment.

- We thank the reviewer for this observation and agree that this would be an interesting hypothesis to test. We plan to look into mechanisms of resistance in future studies.

5. Mechanistically, the authors show marked increase of H2AXphosphorylation when combining ATR and PKMTY1 inhibitors in all tested cell lines, but this increase in not observed when cells are treated

with RP-3500 alone in any of the cells analyzed. Being RP-3500 an inhibitor of ATR, I would expect an increase of DNA damage when used as single treatment. How do the authors explain this observation? Moreover, in Figure 4A it seems that RP-6306 FT282 WT cells display a significant increase of H2AX phosphorylation that is not observed CCNE1/OE FT282 cells. Does it mean that CCNE1 OE reduces DNA damage after inhibition of PKMTY1?. Is this observed in other cell types?

- We thank the reviewer for pointing this out. In the old Figure 4A (Revised Fig. S6A) the RP-3500 and RP-6306 labels were flipped, which explains the confusion and we apologize for this error. With the correct labels we see that indeed RP-6306 induces more γ H2AX in FT282 CCNE1-OE compared to WT as we previously reported. This correction also helps answer the first question regarding the effect of ATRi on DNA damage induction. Using QIBC we do see an induction of γ H2AX with RP-3500 in CCNE1-OE cells, and to a lesser extent WT cells, in both EdU⁺ (Revised Figure S6A) and EdU⁻ (Revised Fig. 4A). Using western blot we also observe a slight induction of γ H2AX in the CCNE1-amp cancer cell lines (Revised Fig. 4C,D).

6. In figure 6I-F, a cleaved caspase.3 demonstrates that combination of the two drugs enhances apoptotic cells death. However, in KLE cells without any treatment there a marked increase of caspase-3 after 48 hours is observed. Although it might be due to unequal protein loading, I think that these western should be improved and additional markers of apoptosis should be shown (annexin V, caspase-9, PARP). I would also recommend showing a immunohistochemistry of cleaved caspase-3 in OVAR3 xenotransplants (as shown for H2AX).

- We have shown the apoptosis, which is Annexin V, in our previous version. We repeated the cleaved caspase-3 and additionally tested the cleaved caspase-9 and cleaved PARP. The combination showed much better effect in inducing caspase protein cleavage, leading to cell apoptosis (Revised Fig. 4H-K).

7. Finally, it is well known that pharmacological inhibitors are not completely specific, and they target other molecules besides the ones that have been designed for. For this reason, it would be interesting to demonstrate that genetic inhibition of ATR and PKMTY1 inhibition has similar effects to those observed with the inhibitors. These experiments wouldn't rule out the possible effects on other targets, but at least they would demonstrate that the anti-cancer effects are caused by inhibition of the expected targets.

- We tested the siRNA of ATR and PKMYT1 to test the off target effect. The combinations of siRNAs showed similar effect as the inhibitors, proving that the inhibitors are effective through inhibiting respective proteins.

References

- 1 Xu, H. *et al.* CCNE1 copy number is a biomarker for response to combination WEE1-ATR inhibition in ovarian and endometrial cancer models. *Cell Rep Med* **2**, 100394, doi:10.1016/j.xcrm.2021.100394 (2021).

- 2 Welcker, M. & Clurman, B. E. FBW7 ubiquitin ligase: a tumour suppressor at the crossroads of cell division, growth and differentiation. *Nat Rev Cancer* **8**, 83-93, doi:10.1038/nrc2290 (2008).
- 3 Yeh, C. H., Bellon, M. & Nicot, C. FBXW7: a critical tumor suppressor of human cancers. *Mol Cancer* **17**, 115, doi:10.1186/s12943-018-0857-2 (2018).
- 4 O'Brien, S. *et al.* FBXW7-loss Sensitizes Cells to ATR Inhibition Through Induced Mitotic Catastrophe. *Cancer Res Commun* **3**, 2596-2607, doi:10.1158/2767-9764.CRC-23-0306 (2023).
- 5 Gallo, D. *et al.* CCNE1 amplification is synthetic lethal with PKMYT1 kinase inhibition. *Nature* **604**, 749-756, doi:10.1038/s41586-022-04638-9 (2022).
- 6 Saldivar, J. C., Cortez, D. & Cimprich, K. A. The essential kinase ATR: ensuring faithful duplication of a challenging genome. *Nat Rev Mol Cell Biol* **18**, 622-636, doi:10.1038/nrm.2017.67 (2017).
- 7 Falchook, G. S. *et al.* A phase Ib study of adavosertib, a selective Wee1 inhibitor, in patients with locally advanced or metastatic solid tumors. *Invest New Drugs* **41**, 493-502, doi:10.1007/s10637-023-01371-6 (2023).
- 8 Dillon, M. T. *et al.* Durable responses to ATR inhibition with ceralasertib in tumors with genomic defects and high inflammation. *J Clin Invest* **134**, doi:10.1172/JCI175369 (2024).
- 9 Yang, L. *et al.* Cyclin-dependent kinase 2 is an ideal target for ovary tumors with elevated cyclin E1 expression. *Oncotarget* **6**, 20801-20812, doi:10.18632/oncotarget.4600 (2015).

Sincerely,

Fiona Simpkins, MD
Hilarie and Mitchell Morgan President's Distinguished Professor in Women's Health
Director of Gynecologic Oncology Clinical & Translational Research
Penn Medicine Health System
Division of Gynecology Oncology
Department of OB-GYN
Perlman Center of Advanced Medicine
3400 Civic Center Blvd, South Tower, Suite 10-176
Philadelphia, PA, 19106
Email: fiona.simpkins@penmedicine.upenn.edu
Office: 215-662-3318
Fax: 215-349-5849

December 5, 2024

Dear Reviewers:

We thank you for your time to review our **revised manuscript**. Your comments have been invaluable in further strengthening this work. After careful revision, we have made additions that we hope will address the remaining comments. Below, we provide a detailed point-by-point response. Reviewers' comments are italicized; bullet points precede our responses in blue.

Point-by-point response

REVIEWER #1:

The authors have substantially improved the manuscript with new experiments though in-depth mechanistic insights are still lacking for this combination.

1) comment on Figure 4G: The gamma-H2AX intensity in the RP-6306 representative image appears higher than in the combination treatment image, which seems inconsistent with the quantification data presented in the bar graph. Please carefully review the representative images and verify the accuracy of the quantification process.

- We thank the reviewer for this observation, and we agree that the representative images do not fully reflect the quantitation presented. In the original analysis we used a scoring algorithm optimized for γ H2AX quantitation using OVCAR3 samples. In the new analysis presented (revised Figure 4G), we have optimized the algorithm by increasing the upper limits of detection for human endometrial cancer PDX samples. New analysis of WU-115 PDX samples now clearly shows a significant increase in 3+ strong positive γ H2AX nuclei in the combo treatment compared to vehicle or RP-6306/RP-3500 single agents. For clarity, we included new representative images with red arrows indicating the 3+ strong positive γ H2AX nuclei that the improved algorithm is scoring. Finally, we have updated the methods section to reflect that different algorithms were used to score γ H2AX IHC in OVCAR3 and WU-115.

REVIEWER #3:

Although the authors have addressed and solved most of my concerns, there still are two important points that need to be further clarified.

1. In my first review I expressed my main concern about the significance of the results:

“My main and general concern is related to the clinical relevance of author’s findings. Although CCNE1 overexpressing EMCA account for 5-6% of all endometrial neoplasia, they are high grade cancer, with aggressive behavior and metastatic potential. From this point of view, the proposed combinatorial treatment can be a new therapeutic approach for these patients. However, metastatic outgrowth is the main responsible for poor prognosis of these type of tumors. For this reason, It would be very interesting to analyze the effects of RP- 6306 and RP-3500 combination in metastases of EMCA and OVCA.”

The authors answered my question as follows.

• Metastatic ovarian and endometrial cancers are indeed a major clinical challenge us physicians face when treating these patients. Two of 3 PDX models tested (WO-77, WO-19) developed tumor metastasis and all develop ascites which were significantly reduced with the combination making this combination clinically relevant.

However, I am unable to find the results regarding the effects of drug combination on metastatic outgrowth of the above-mentioned PDX. Could the authors clarify in which figures are shown the effects on metastatic spreads?

Moreover, I am unable to find the word metastasis in any section of the results nor in the figure legends, where are these results explained?

- We thank the reviewer for this observation, and we have added data showing drug control of metastases and ascites in our PDX models to the main figure and supplemental figure, respectively. We noted a significant decrease in metastases with combination treatment compared to control in our two ovarian cancer PDX models (WO-19: $p=0.012$ and WO-77: $p<0.0001$). While there was not a significant decrease in the endometrial cancer model (WU-115), there was a trend toward decreased metastases with the combination ($p=0.079$). Regarding ascites, there was a significant decrease in ascites at the end of the experiment in the WO-77 model. The WO-19 model did not develop ascites in any arm. For the endometrial cancer model, WU-115, there were a few mice in the control and monotherapy arms that developed small to medium ascites, however none in the combination groups developed ascites, although this difference was nonsignificant. Please see revised Fig 3 and Sup Fig 4.

2. To my previous question:

“In Figure 3, the authors clearly show a synergistic effect ofc in PDX models of OVCA and EMCA. In all PDX, combination therapy significantly delays progression of disease. However, after several weeks of treatment (depending on PDX), tumors start to grow again reaching volumes above 1000 mm. This It is probably caused by acquired resistance to treatment. Theoretically, clones lacking CCNE1 amplification or clone, but other molecular alterations would be responsible for such resistance. Alternatively, clones of cells containing CCNE1 AMP , acquire other mutations conferring resistance to treatment. Therefore, from the clinical point of view, It would be very interesting to see whether resistant tumors that progress after treatment display CCNE1 amplification. At the molecular level, it would be

interesting to see whether these tumors activate apoptosis and H2AX during regression period, but no when they acquire resistance to treatment.

The authors answered my question as follows.

• We thank the reviewer for this observation and agree that this would be an interesting hypothesis to test. We plan to look into mechanisms of resistance in future studies.

I understand the authors will study in depth the molecular mechanisms of resistance and these results are beyond the scope of the present work. However, I think that these observations about resistance should be properly stated and discussed.

- We have added a statement in the results section acknowledging the observed resistance in the combination treatment groups and added statements on potential mechanisms in the discussion.

Sincerely,

Fiona Simpkins, MD
Hilarie and Mitchell Morgan President's Distinguished Professor in Women's Health
Director of Gynecologic Oncology Clinical & Translational Research
Penn Medicine Health System
Division of Gynecology Oncology
Department of OB-GYN
Perlman Center of Advanced Medicine
3400 Civic Center Blvd, South Tower, Suite 10-176
Philadelphia, PA, 19106
Email: fiona.simpkins@pennmedicine.upenn.edu
Office: 215-662-3318
Fax: 215-349-5849